# Policy Optimization for Continuous Reinforcement Learning

**Hanyang Zhao**
Columbia University
hz2684@columbia.edu

**Wenpin Tang**
Columbia University
wt2319@columbia.edu

**David D. Yao**
Columbia University
yao@columbia.edu

## Abstract

We study reinforcement learning (RL) in the setting of continuous time and space, for an infinite horizon with a discounted objective and the underlying dynamics driven by a stochastic differential equation. Built upon recent advances in the continuous approach to RL, we develop a notion of occupation time (specifically for a discounted objective), and show how it can be effectively used to derive performance-difference and local-approximation formulas. We further extend these results to illustrate their applications in the PG (policy gradient) and TRPO/PPO (trust region policy optimization/ proximal policy optimization) methods, which have been familiar and powerful tools in the discrete RL setting but under-developed in continuous RL. Through numerical experiments, we demonstrate the effectiveness and advantages of our approach.

## 1 Introduction

Reinforcement Learning (RL, [52]) has been successfully applied to wide-ranging domains in the past decade, including achieving superhuman performance in games like Atari and Go [35, 48, 49], enhancing Large Language Models using human feedback [8, 10], and showing potentials in improving traditional model-based decisions in healthcare, inventory management, and finance [9, 30, 33]. Most existing works, including all references cited above, are formulated and solved as discrete-time sequential optimization problems such as Markov decision processes (MDPs, [42]). Yet in many applications, agents may need to monitor and interact with the random environment at an ultra-high frequency (e.g., autonomous driving, robot navigation, and high-frequency stock trading), which calls for a *continuous-time/space* approach.

Recent years have witnessed a fast growing body of research that has extended the frontiers of continuous RL in several important directions including, for instance, modeling the noise or randomness in the environment dynamics as following a stochastic differential equation (SDE), and incorporating an entropy-based regularizer into the objective function [58] to facilitate the exploration-exploitation tradeoff; designing model-free methods and algorithms, along with applications to portfolio optimization [18, 20, 21, 22]; studying regret bounds [5, 54], and so forth.

In this paper, we continue the above trend in continuous RL, focusing on an infinite horizon formulation with a discounted objective and the underlying dynamics driven by an SDE [24, 39]. We are specifically motivated by the following two questions.

(**Q1**) The visitation frequency in MDP (with a discounted objective) is defined as: $\rho(s) = \sum_{t=0}^{\infty} \gamma^t \cdot \mathbb{P}(Y_t = s)$, where $\{Y_t\}$ is a Markov chain with state space $\mathcal{S} := \{s\}$, and $\gamma \in (0, 1)$ is a discount factor. It plays an important role in many RL algorithms for MDP. So, a natural question is, what is the continuous counterpart of $\rho(s)$?

(**Q2**) For continuous RL, how can we characterize the difference in performance between two policies? In particular, can we derive performance-difference formulas similar to those in the MDP

37th Conference on Neural Information Processing Systems (NeurIPS 2023).

case [23, 45]? Can we adapt and apply the ideas and tools of the efficient policy optimization methods (e.g., [45, 47]) to the continuous RL setting?

**Main contributions**. We provide a unified theory/framework for policy optimization in continuous time and space. Specifically, we have addressed the above two questions **(Q1)** and **(Q2)** by developing the notion of *occupation time/measure*, specifically for a discounted objective, and focusing on its associated *q-value*. Based on these two quantities, we derive the performance-difference formula for continuous RL by means of perturbation analysis. Leveraging the performance-difference formula, we develop the continuous counterparts of the policy gradient (PG, [53]) and also propose the local approximation for the performance metric, for which we derive a bound on it and allow the development of a minorization-majorization (MM) algorithm. We further develop the continuous counterparts of trust region policy optimization/ proximal policy optimization (TRPO/PPO) methods in [45, 47], which have been familiar and powerful tools in the discrete RL setting but under-developed in continuous RL, as approximations to the previous algorithms. (What is worth mentioning is that these policy optimization algorithms do not require any *a priori* discretization of time and space.) Through numerical examples we show the convergence of these algorithms when applied to certain stochastic control tasks in continuous time and space.

**Organization of the paper**. In Section 2 we present the continuous RL formulation and develop necessary tools. The main results, the performance-difference formula (Theorem 2) and the bound (Theorem 5) are provided in Section 3. In Section 4 we propose two algorithms, *policy gradient with random rollout* and *PPO with adaptive penalty*, based on our analyses and theoretical results; and illustrate their performance via numerical experiments. Concluding remarks are summarized in Section 5.

**Related works**. One line of research on continuous RL focuses on modeling the underlying dynamics as a deterministic system, typically following a deterministic ordinary differential equation. Several papers [9, 37, 44] solve the problems via *a priori* discretization in either time or space; [13] develops a framework to apply the temporal difference to the continuous setting, and proposes algorithms that combine value iteration or advantage update as in [3, 4, 6] to avoid explicit discretization; [37] further investigates policy gradient methods, followed by more recent studies on model-free continuous RL methods [25, 29, 57] or model-based ones [14]; [55] studies the sensitivity of existing off-policy algorithms along with advantage updating to propose continuous RL algorithms that are robust to time discretization.

The formulation of continuous RL in a stochastic setting (i.e., with the state process driven by an SDE), can be traced back to [38], which however provides no data-driven solution. Recently, [58] develops an exploratory control model for the continuous RL. Built upon this approach and for a finite-horizon objective, [20] studies policy evaluation, and [21] policy gradient. Furthermore, [22] brings forth the notion of $q$-value, which leads to a continuous analogue of $Q$-learning. Also worth noting is [2, 28], which studies RL in the mean-field regime where continuous-time processes occur in the limit, and [16] extends the study to jump-diffusion processes.

In discrete-time MDPs, the body of research on bounding the performance difference between two policies also relates to our work: [1, 45] develop a policy improvement bound for the discounted total reward; [12, 64] studies the long-run average reward, and [11] proposes a bound that is continuous with respect to the discount factor.

**Notation**. For a measurable set $\mathcal{A}$, denote $\mathcal{P}(\mathcal{A})$ for the set of probability distributions over $\mathcal{A}$. For a vector $x$, denote by $||x||_2$ the Euclidean norm of $x$. For a matrix $A$, denote by $||A||_F$ the Frobenius norm of $A$, and $A^2 := AA^\top$ where $A^\top$ is the transpose of $A$. For a positive-definite matrix $A$, denote by $A^{\frac{1}{2}}$ the square root matrix of $A$. For $A$, $B$ two matrices of the same size, denote by $A \circ B$ the inner product of $A$ and $B$. For a function $f$ on an Euclidean space, $\nabla f$ (resp. $\nabla^2 f$) denotes the gradient (resp. the Hessian) of $f$. For two distributions $P, Q \in \mathcal{P}(A)$, denote by $W_2(P, Q)$ the Wasserstein-2 distance (or Quadratic Wasserstein distance) between $P$ and $Q$:

$$W_2(P,Q) = \left( \inf_{\gamma \in \Gamma(P,Q)} \mathbf{E}_{(x,y)\sim\gamma} \|x-y\|^2 \right)^{1/2},$$

where $\Gamma(P,Q)$ is the set of all couplings of $P$ and $Q$; and denote by $D_{\mathrm{KL}}(P||Q)$ the KL-divergence between $P$ and $Q$: $D_{\mathrm{KL}}(P||Q) = \int p(x) \log \left( \frac{p(x)}{q(x)} \right) dx$, in which $p$ and $q$ denote the probability densities of $P$ and $Q$.

## 2 Formulation and Preliminaries

**Continuous RL**. We start with a quick formulation of the continuous RL, based on the same modeling framework as in [58]. Assume that the state space is $\mathbb{R}^n$, and denote by $\mathcal{A}$ the action space. Let $\pi(\cdot \mid x) \in \mathcal{P}(\mathcal{A})$ be a (state) feedback policy given the state $x \in \mathbb{R}^n$. A continuous RL problem is formulated by a distributional (or relaxed) control approach [62], which is motivated by the trial and error process in RL. The state dynamics $(X_s^a, \ s \geq 0)$ is governed by the Itô process:

$$\mathrm{d}X_s^a = b\left(X_s^a, a_s\right)\mathrm{d}s + \sigma\left(X_s^a, a_s\right)\mathrm{d}B_s, \quad X_0^a \sim \mu \in \mathcal{P}(\mathbb{R}^n), \tag{1}$$

where $(B_t, \ t \geq 0)$ is the $m$-dimensional Brownian motion, $b : \mathbb{R}^n \times \mathcal{A} \mapsto \mathbb{R}^n$, $\sigma : \mathbb{R}^n \times \mathcal{A} \mapsto \mathbb{R}^{n \times m}$, and the action $a_s$ is generated from the distribution $\pi\left(\cdot \mid X_s^a\right)$ by *external randomization*. To avoid technical difficulties, we assume that the stochastic processes (1) (and (3), (9) below) are well-defined, see [24, Section 5.3] or [50, Chapter 6] for background.

From now on, write $(X_s^\pi, a_s^\pi)$ for the state and action at time $s$ given by the process (1) under the policy $\pi = \{\pi(\cdot \mid x) \in \mathcal{P}(\mathcal{A}) : x \in \mathbb{R}^n\}$. The goal here is to find the optimal feedback policy $\pi^*$ that maximizes the expected discounted reward over an infinite time horizon:

$$V^* := \max_\pi \mathbb{E}\left[\int_0^{+\infty} e^{-\beta s} \left[r\left(X_s^\pi, a_s^\pi\right) + \gamma p\left(X_s^\pi, a_s^\pi, \pi\left(\cdot \mid X_s^\pi\right)\right)\right]\mathrm{d}s \mid X_0^\pi \sim \mu\right], \tag{2}$$

where $r : \mathbb{R}^n \times \mathcal{A} \mapsto \mathbb{R}^+$ is the running reward of the current state and action $(X_s^\pi, a_s^\pi)$; $p : \mathbb{R}^n \times \mathcal{A} \times \mathcal{P}(\mathcal{A}) \mapsto \mathbb{R}$ is a regularizer which facilitates exploration (e.g., in [58], $p$ is taken as the differential entropy defined by $p(x, a, \pi(\cdot)) = -\log \pi(a)$); $\gamma \geq 0$ is a weight parameter on exploration (also known as the "temperature" parameter); and $\beta > 0$ is a discount factor that measures the time-depreciation of the objective value (or the impatience level of the agent).

**Performance metric**. A standard approach to solving the problem in (2) is to find a sequence of policies $\pi_k = \{\pi_k(\cdot \mid x) : x \in \mathbb{R}^n\}$, $k = 1, 2, \ldots$ such that the value functions following the policies $\pi_k$ will converge to $V^*$, or be at least increasing in $k$, i.e., demonstrating policy improvement.

Given a policy $\pi(\cdot)$, let $\tilde{b}(x, \pi(\cdot)) := \int_{\mathcal{A}} b(x, a)\pi(a)\mathrm{d}a$ and $\tilde{\sigma}(x, \pi(\cdot)) := \left(\int_{\mathcal{A}} \sigma^2(x, a)\pi(a)\mathrm{d}a\right)^{\frac{1}{2}}$. Assume (for technical purpose) that $\tilde{\sigma}(x, \pi(\cdot))$ is positive definite for every $x \in \mathbb{R}^n$. It is sometimes more convenient to consider the following equivalent SDE representation of (1):

$$\mathrm{d}\tilde{X}_s = \tilde{b}\left(\tilde{X}_s, \pi(\cdot \mid \tilde{X}_s)\right)\mathrm{d}s + \tilde{\sigma}\left(\tilde{X}_s, \pi(\cdot \mid \tilde{X}_s)\right)\mathrm{d}\tilde{B}_s, \quad \tilde{X}_0 \sim \mu, \tag{3}$$

in the sense that there exists a probability measure $\tilde{\mathbb{P}}$ which supports the $m$-dimensional Brownian motion $(\tilde{B}_s, \ s \geq 0)$, and for each $s \geq 0$, the distribution of $\tilde{X}_s$ under $\tilde{\mathbb{P}}$ agrees with that of $X_s$ under $\mathbb{P}$ defined by (1), see Appendix A. Note that the dynamics in (3) does not require external randomization. Also set $\tilde{r}(x, \pi) := \int_{\mathcal{A}} r(x, a)\pi(a)\mathrm{d}a$ and $\tilde{p}(x, \pi) := \int_{\mathcal{A}} p(x, a, \pi)\pi(a)\mathrm{d}a$. We formally define the (state) value function given the feedback policy $\{\pi(\cdot \mid x) : x \in \mathbb{R}^n\}$ by

$$\begin{aligned} V(x; \pi) &:= \mathbb{E}\left[\int_0^{+\infty} e^{-\beta s}\left[r\left(X_s^\pi, a_s^\pi\right) + \gamma p\left(X_s^\pi, a_s^\pi, \pi\left(\cdot \mid X_s^\pi\right)\right)\right]\mathrm{d}s \mid X_0^\pi = x\right] \\ &= \mathbb{E}\left[\int_0^{\infty} e^{-\beta s}\left[\tilde{r}\left(\tilde{X}_s^\pi, \pi(\cdot \mid \tilde{X}_s^\pi)\right) + \gamma\tilde{p}\left(\tilde{X}_s^\pi, \pi(\cdot \mid \tilde{X}_s^\pi)\right)\right]\mathrm{d}s \mid \tilde{X}_0^\pi = x\right], \end{aligned} \tag{4}$$

which, under suitable conditions on model parameters $(b, \sigma, r, p)$ and the policy $\pi$, is characterized by the Hamilton-Jacobi equation (see [21, 56]):

$$\beta V(x; \pi) - \tilde{b}(x, \pi) \cdot \nabla V(x; \pi) - \frac{1}{2}\tilde{\sigma}^2(x, \pi) \circ \nabla^2 V(x; \pi) - \tilde{r}(x, \pi) - \gamma\tilde{p}(x, \pi) = 0. \tag{5}$$

More technical details regarding the above formulation are spelled out in the Appendix.

We can now define the performance metric as follows:

$$\eta(\pi) := \int_{\mathbb{R}^n} V(x; \pi)\mu(dx), \tag{6}$$

so $V^* = \max_\pi \eta(\pi)$. The main task of the continuous RL is to approximate $\max_\pi \eta(\pi)$ by constructing a sequence of policies $\pi_k$, $k = 1, 2, \ldots$ recursively such that $\eta(\pi_k)$ is non-decreasing.

**Policy evaluation**. Let's first recall a general approach in [20], which can be used to learn the state value function in (4) or the performance metric in (6) for a given policy $\pi$. The idea is that for any $T > 0$ and a suitable test process $(\xi_t,\ t \geq 0)$,

$$\mathbb{E} \int_0^T \xi_t \left[ \mathrm{d}V\left(X_t^\pi; \pi\right) + r\left(X_t^\pi, a_t^\pi\right) \mathrm{d}t + \gamma p\left(X_t^\pi, a_t^\pi, \pi\left(\cdot \mid X_t^\pi\right)\right) \mathrm{d}t - \beta V\left(X_t^\pi; \pi\right) \mathrm{d}t \right] = 0. \quad (7)$$

If we parameterize $V(x; \pi) = V^\phi(x)$ and choose the special test function $\xi_t = \frac{\partial V^\phi(X_t^\pi)}{\partial \phi}$, stochastic approximation leads to the online update:

$$\phi \leftarrow \phi + \alpha \frac{\partial V^\phi\left(X_t^\pi\right)}{\partial \phi} [\mathrm{d}V^\phi\left(X_t^\pi\right) + r\left(X_t^\pi, a_t^\pi\right) \mathrm{d}t + \gamma p\left(X_t^\pi, a_t^\pi, \pi\left(\cdot \mid X_t^\pi\right)\right) \mathrm{d}t - \beta V^\phi\left(X_t^\pi\right) \mathrm{d}t],$$
$$(8)$$

where $\alpha > 0$ is the learning rate. This recovers the mean-squared TD error (MSTDE) method for policy evaluation in the discrete RL [51].

**$q$-value**. The Q-value function [61] and the advantage function [3, 34] in discrete-time MDPs play a critical role in reinforcement learning theory and algorithms. However, as pointed out in [55], these concepts will not apply when the time interval shrinks to $0$ (as in the continuous setting). To derive algorithms that fit the need of a continuous stochastic environment, [55, 22] proposed the advantage *rate* function. Namely, given a policy $\pi$ and $(t, x, a) \in [0, \infty) \times \mathbb{R}^n \times \mathcal{A}$, consider a "perturbed" policy $\hat{\pi}$ as follows. It takes the action $a \in \mathcal{A}$ on $[t, t + \Delta t]$ where $\Delta t > 0$, and then follows $\pi$ on $[t + \Delta t, \infty)$. The corresponding state process $X^{\hat{\pi}}$ given $X_t^{\hat{\pi}} = x$ is broken into two pieces. On $[t, t + \Delta t)$, it is $X^a$ which is the solution to

$$\mathrm{d}X_s^a = b\left(X_s^a, a\right) \mathrm{d}s + \sigma\left(X_s^a, a\right) \mathrm{d}B_s, s \in [t, t + \Delta t), \quad X_t^a = x, \quad (9)$$

while on $[t + \Delta t, \infty)$, it is $X^\pi$ following (3) with the initial time-state pair $\left(t + \Delta t, X_{t+\Delta t}^a\right)$. With $\Delta t > 0$ as time discretization, the generalization of the conventional Q-function can be expressed as

$$Q_{\Delta t}(x, a; \pi) = V(x; \pi) + \left[ \mathcal{H}^a \left( x, \frac{\partial V}{\partial x}(x; \pi), \frac{\partial^2 V}{\partial x^2}(x; \pi) \right) - \beta V(x; \pi) \right] \Delta t + o(\Delta t). \quad (10)$$

where $\mathcal{H}^a(x, y, A) := b(x, a) \cdot y + \frac{1}{2}\sigma^2(x, a) \circ A + r(x, a)$ is the (generalized) Hamilton function in stochastic control theory [62]. This motivates the following definition.

**Definition 1.** *[22] For a given policy $\pi \in \Pi$ and $(x, a) \in \mathbb{R}^n \times \mathcal{A}$, define the q-value as*

$$q(x, a; \pi) := \lim_{\Delta t \to 0} \frac{Q_{\Delta t}(x, a; \pi) - V(x; \pi)}{\Delta t} = \mathcal{H}^a \left( x, \frac{\partial V}{\partial x}(x; \pi), \frac{\partial^2 V}{\partial x^2}(x; \pi) \right) - \beta V(x; \pi),$$
$$(11)$$

*which represents the instantaneous advantage rate of an action in a given state under a given policy.*

## 3 Main Results

This section is concerned with theoretical developments. In Section 3.1, we define the (discounted) occupation time/measure which is the continuous analog of visitation frequency in MDPs. It is crucial in deriving the performance-difference formula in Section 3.2, which spins off two different algorithms – policy gradient and TRPO/PPO. In Section 3.3, we propose a local approximation for the performance metric, and derive a bound from which an MM algorithm is constructed.

### 3.1 Discounted Occupation Time

Here we first provide an answer to (**Q1**) by defining the notion of *discounted occupation time* for the continuous RL.

**Definition 2.** *Let $X = (X_t,\ t \geq 0)$ be governed by the SDE (3), and assume that it has a probability density function $p^\pi(\cdot, t)$ at each time $t$. For each $x \in \mathbb{R}^n$ and $t \geq 0$, define the $\beta$-discounted occupation time of $X$ at the state $x$ by*

$$d_\mu^\pi(x) := \int_0^\infty e^{-\beta s} p^\pi(x, s) ds. \quad (12)$$

*So $d_\mu^\pi(\cdot)$ induces a finite measure on $\mathbb{R}^n$ with a total mass of $\beta^{-1}$, which we call the discounted occupation measure.*

In probability theory, the definition in (12) is referred to as the $\beta$-potential of $X$, which gives discounted visitation frequencies of the state process. We record the following result, which will be useful in the derivation of the performance-difference formula. It is a consequence of the occupation time formula [41, 43].

**Lemma 1.** *Under the conditions in 2, we have* $\mathbb{E} \int_0^\infty e^{-\beta s} \varphi(X_s) \, \mathrm{d}s = \int_{\mathbb{R}^n} d_\mu^\pi(x) \varphi(x) \mathrm{d}x$, *for any measurable function* $\varphi : \mathbb{R}^n \mapsto \mathbb{R}_+$ *for which the expectation exists.*

### 3.2 Performance-Difference Formula

We are now ready to answer (**Q2**) by deriving the performance-difference formula between two policies in terms of the discounted occupation time in (12) and the $q$-values in (11).

**Theorem 2.** *Given two feedback policies* $\hat{\pi} = \{\hat{\pi}(\cdot \mid x) : x \in \mathbb{R}^n\}$ *and* $\pi = \{\pi(\cdot \mid x) : x \in \mathbb{R}^n\}$, *we have:*

$$\eta(\hat{\pi}) - \eta(\pi) = \int_{\mathbb{R}^n} d_\mu^{\hat{\pi}}(x) \left[ \int_{\mathcal{A}} \hat{\pi}(a \mid x) \left( q(x, a; \pi) + \gamma p(x, a, \hat{\pi}) \right) \mathrm{d}a \right] \mathrm{d}x. \tag{13}$$

*Proof sketch.* The full proof is detailed in Appendix B.1. The essence of the proof is to use the perturbation theory and properties of the discounted occupation time. Define an operator $\mathcal{L}^\pi : C^2(\mathbb{R}^n) \mapsto C(\mathbb{R}^n)$ associated with the diffusion process as:

$$(\mathcal{L}^\pi \varphi)(x) := -\beta \varphi(x) + \tilde{b}(x, \pi) \cdot \nabla \varphi(x) + \frac{1}{2} \tilde{\sigma}(x, \pi)^2 \circ \nabla^2 \varphi(x). \tag{14}$$

Then the Hamilton-Jacobi equation that characterizes the state value function can be expressed as:

$$-\mathcal{L}^\pi V(x; \pi) = \tilde{r}(x, \pi) + \gamma \tilde{p}(x, \pi) \tag{15}$$

Note that for any $\varphi \in C^2(\mathbb{R}^n)$, we have $\int_{\mathbb{R}^n} d_\mu^\pi(y)(-\mathcal{L}^\pi \varphi)(y) \mathrm{d}y = \int_{\mathbb{R}^n} \varphi(y) \mu(\mathrm{d}y)$. This allows us to express the performance difference in model-dynamics related terms:

$$\eta(\hat{\pi}) - \eta(\pi) = \int_{\mathbb{R}} d_\mu^{\hat{\pi}}(y) \left[ (\mathcal{L}^{\hat{\pi}} - \mathcal{L}^\pi) V(y; \pi) + \tilde{r}(y, \hat{\pi}) + \gamma \tilde{p}(y, \hat{\pi}) - \tilde{r}(y, \pi) - \gamma \tilde{p}(y, \pi) \right] \mathrm{d}y. \tag{16}$$

What remains is to reduce the above to the desired result in (13). $\qquad\square$

As discussed in Section 2, our main task is to construct (algorithmically) a sequence of policies $\pi_k$ along which the performance improves. Here we illustrate how some well known approaches of policy improvement (from $\pi$ to $\hat{\pi}$) are instances of the performance difference formula (13).

$q$**-learning and soft** $q$**-learning**. Since $d_\mu^{\hat{\pi}} \geq 0$, we only need to ensure that for all $x \in \mathbb{R}^n$, $\int_{\mathcal{A}} \hat{\pi}(a \mid x)(q(x, a; \pi) + \gamma p(x, a, \hat{\pi})) \mathrm{d}a \geq 0$. This boils down to the problem that for any $x$, find $v \equiv \hat{\pi}(\cdot|x)$ to maximize

$$\int_{\mathcal{A}} v(a) \left( q(x, a; \pi) + \gamma p(x, a, v) \right) \mathrm{d}a. \tag{17}$$

There are two special cases:

*(i)* If $p(x, a, v) \equiv 0$, then $v = \delta(a^*)$ where $a^* = \arg\max_a q(x, a, \pi)$. This is essentially the counterpart of $Q$-Learning [61] in the discrete time, which we call $q$-learning.

*(ii)* If $p(x, a, v) = -\log(v(a))$, this is known as the entropy regularizer [17, 59]. Concretely, we need to solve

$$\max_{v \in \mathcal{P}(\mathcal{A})} \int_{\mathcal{A}} v(a) \left( q(x, a; \pi) - \gamma \log v(a) \right) \mathrm{d}a. \tag{18}$$

which has a closed form solution with $v^*(a) \propto \exp(\frac{q(x, a, \pi)}{\gamma})$, i.e. $v^*$ is the Boltzmann policy for $q$-functions. This is a "soft" (*à la* [17]) version of the $q$-learning mentioned above.

**Policy gradient**. We may use function approximations to $\pi$ by a parametric family $\pi^\theta$, with $\theta \in \Theta \subseteq \mathbb{R}^L$. For simplicity, we write $d_\mu^{\theta_0}$ (resp. $\eta(\theta)$) for $d_\mu^{\pi^{\theta_0}}$ (resp. $\eta(\pi^\theta)$). Setting $\hat{\pi} = \pi^\theta$ and $\pi = \pi^{\theta_0}$ in (13) and taking derivative with respect to $\theta$ on both sides, we get the following result.

**Theorem 3** (Policy Gradient). *The policy gradient at $\pi^{\theta_0}$ is:*

$$\nabla_\theta \eta(\theta) \mid_{\theta=\theta_0} = \frac{1}{\beta} \mathbb{E}_{(x,a)} \left[ \nabla_\theta \log \left( \pi^\theta(a \mid x) \right) \left( q(x,a;\pi^{\theta_0}) + \gamma p(x,a,\pi^{\theta_0}) \right) + \gamma \nabla_\theta p(x,a,\pi^\theta) \right],$$

(19)

*where the expectation is w.r.t. $(x,a) \sim (\beta d_\mu^{\theta_0}, \pi^{\theta_0})$, meaning $x \sim \beta d_\mu^{\theta_0}(\cdot)$ and then $a \sim \pi^{\theta_0}(\cdot \mid x)$.*

The above formula is indeed the continuous analogue to the well-known PG formula (without regularization) in the MDP setting, where $\nabla_\theta \eta(\theta) \mid_{\theta=\theta_0} = \frac{1}{\beta} \mathbb{E}_{(x,a)} \left[ \nabla_\theta \log \left( \pi^\theta(a \mid x) \right) A(x,a;\pi^{\theta_0}) \right]$ ([53]), with $A$ denoting the advantage function. Specifically, as a comparison, the formula in (19) replaces the visitation frequency by the occupation time, and the advantage function by the $q$-function, while keeping the same score function $\nabla_\theta \log \left( \pi^\theta(a \mid x) \right)$.

### 3.3 Continuous TRPO/PPO

Leveraging the performance-difference formula derived above, we can now move on to spell out the continuous counterpart of TRPO and PPO originally developed in [45, 47] for the discrete RL.

**Local approximation function**: Given a feedback policy $\hat{\pi}$, we define the *local approximation function* to $\eta(\hat{\pi})$ by

$$L^\pi(\hat{\pi}) = \eta(\pi) + \int_{\mathbb{R}^n} d_\mu^\pi(x) \left[ \int_{\mathcal{A}} \hat{\pi}(a \mid x) \left( q(x,a;\pi) + p(x,a,\hat{\pi}) \right) da \right] dx. \tag{20}$$

Comparing (20) to the formula (13), we see that the difference is to replace $d_\mu^{\hat{\pi}}(s)$ with $d_\mu^\pi(s)$. Observe that

$$\text{(i) } L^\pi(\pi) = \eta(\pi), \quad \text{(ii) } \nabla_\theta \left( \eta(\pi^\theta) \right) \mid_{\theta=\theta_0} = \nabla_\theta \left( L^{\pi^{\theta_0}}(\pi^\theta) \right) \mid_{\theta=\theta_0},$$

i.e. the local approximation function and the true performance objective share the same value and the same gradient with respect to the policy parameters. Thus, the local approximation function can be regarded as the first order approximation to the performance metric. Furthermore, similar to [23, 45], we can apply simulation methods to evaluate the local approximation function only using the data generated from the current policy $\pi$:

$$L^\pi(\hat{\pi}) = \eta(\pi) + \frac{1}{\beta} \mathbb{E}_{(x,a)\sim(\beta d_\mu^\pi, \pi(\cdot \mid x))} \left[ \frac{\hat{\pi}(a \mid x)}{\pi(a \mid x)} \left( q(x,a;\pi) + \gamma p(x,a,\hat{\pi}) \right) \right]. \tag{21}$$

Next, we provide analysis and bounds on the gap $\eta(\hat{\pi}) - L^\pi(\hat{\pi})$, which can then be used to ensure policy improvement (similar to approaches in [45, 64] for discounted/average reward MDP). First, we need some technical conditions on the model dynamics.

**Assumption 1.** *Assume the following conditions for the state dynamics hold true:*
*(i) Global boundedness: There exists $0 < \sigma_0 \leq \bar{\sigma}_0$ such that $\sigma_0^2 \cdot I \leq \tilde{\sigma}^2(x,a) \leq \bar{\sigma}_0^2 \cdot I$ for all $x, a$;*
*(ii) Uniformly Lipschitz: There exists $C_{\tilde{\sigma}} > 0$ such that $\|\tilde{\sigma}(x,\pi) - \tilde{\sigma}(x',\pi)\|_F \leq C_{\tilde{\sigma}} \|x - x'\|_2$ for all $\pi$ and $x, x'$;*
*(iii) Monotonicity (for drift) or growth condition:*
*There exists $C_{\tilde{b}} > 0$ such that $(x - x')^\top \left( \tilde{b}(x,\pi) - \tilde{b}(x',\pi) \right) \leq C_{\tilde{b}} \|x - x'\|_2^2$ for all $\pi$ and $x, x'$.*

The following lemma provides a Wasserstein-2 bound between the discounted occupation measures $d_\mu^\pi(\cdot)$ and $d_\mu^{\hat{\pi}}(\cdot)$ for two policies $\pi$ and $\pi'$.

**Lemma 4.** *Let $\pi, \pi'$ be two feedback policies, and suppose the conditions in Assumption 1 hold. Define $C_{\tilde{b},\tilde{\sigma}} := 2C_{\tilde{b}} + 1 + 2C_{\tilde{\sigma}}^2$ and $C = \sup_{x,a} |b(x,a)|^2 + \frac{n\bar{\sigma}_0^2}{2\sigma_0^2}$. (Recall $n$ is the dimension of the state.) Assume further that $\beta > C_{\tilde{b},\tilde{\sigma}}$ and $C < \infty$. Then there is the bound*

$$W_2 \left( \beta d_\mu^{\hat{\pi}}, \beta d_\mu^\pi \right) \leq \frac{C}{\beta C_{\tilde{b},\tilde{\sigma}}(\beta - C_{\tilde{b},\tilde{\sigma}})} \cdot \max \left( \sup_x \|\hat{\pi}(\cdot \mid x) - \pi(\cdot \mid x)\|_1, \sup_x \|\hat{\pi}(\cdot \mid x) - \pi(\cdot \mid x)\|_1^{\frac{1}{2}} \right).$$

(22)

(The proof is deferred to Appendix B.3.) To derive a performance difference bound, define the Sobolev semi-norm as $K := \|f\|_{\dot{H}^1} := \left( \int_{\mathbb{R}^n} |\nabla f(x)|^2 dx \right)^{\frac{1}{2}}$, and its dual norm $\| \cdot \|_{\dot{H}^{-1}}$ as $\|\mu\|_{\dot{H}^{-1}} =$

$\sup\{|\langle g,\mu\rangle|\mid \|g\|_{\dot{H}^1}\leq 1\}$. [32, 40] show the equivalence of this dual norm $\|\mu-v\|_{\dot{H}^{-1}}$ to the Wasserstein-2 distance $W_2(\mu,v)$ for any probability measure $\mu$ and $v$. Combining this fact with Lemma 4 yields the following result.

**Theorem 5.** *Suppose the conditions in Lemma 4 hold, and further assume that $d_\mu^{\hat\pi}(x), d_\mu^\pi(x)$ are upper bounded by $M$ for all $x \in \mathbb{R}^n$. Define $K := \|f\|_{\dot{H}^1}$ with $f(x;\pi,\hat\pi) := \int_{\mathcal{A}} \hat\pi(a \mid x)\,(q(x,a;\pi)+p(x,a,\hat\pi))\,\mathrm{d}a$ and $C(\mu,\pi,\hat\pi) := \frac{\sqrt{M}K}{2\beta^2 C_{\tilde{b},\tilde\sigma}(\beta-C_{\tilde{b},\tilde\sigma})}\left(\sup_{x,a}\|b(x,a)\|^2+\frac{n\tilde\sigma_0^2}{2\sigma_0^2}\right)$. Assuming $C(\mu,\pi,\hat\pi)<\infty$, we have $\eta(\hat\pi)\geq \underline{L}^\pi(\hat\pi)$, where*

$$\underline{L}^\pi(\hat\pi) := L^\pi(\hat\pi) - C(\mu,\pi,\hat\pi)\cdot\max\left(\sup_x D_{\mathrm{KL}}(\hat\pi(\cdot|x)\|\pi(\cdot|x)),\sup_x\sqrt{D_{\mathrm{KL}}(\hat\pi(\cdot|x)\|\pi(\cdot|x))}\right). \tag{23}$$

Proof is given in Appendix B. By Theorem 5, we can use the minorization-maximization (MM) algorithm in [19, 23, 26], where $\underline{L}^\pi(\hat\pi)$ is taken as the surrogate function for $\eta(\pi)$. Specifically, given the policy $\pi_k$, if we can indeed solve the optimization problem $\max_{\hat\pi}\underline{L}^{\pi_k}(\hat\pi)$, and designate its solution as $\pi_{k+1}$. Then, we have

$$\eta(\pi_{k+1})\geq \underline{L}^{\pi_k}(\pi_{k+1})\geq \underline{L}^{\pi_k}(\pi_k)=\eta(\pi_k) \tag{24}$$

i.e., a guaranteed performance improvement. See also [26, Chapter 7] and [27] for the (global) convergence analysis of the MM algorithm (which exceeds the scope of this work). However, in general this optimization problem is not easy to solve directly since $C(\mu,\pi,\hat\pi)$ is unknown because of the unknown underlying dynamics, and we may also have to work with sample based estimates of the approximation functions. In the spirit of [45, 47], we provide algorithms in the next section that can be practically implemented by incorporating an adaptive penalty constant $C_{\text{penalty}}$ as an alternative to $C(\mu,\pi,\hat\pi)$. Consequently, the resulting algorithms may no longer strictly preserve the increasing performance of $\eta$ at each iteration, but overall increasing *trend* will be clear (as demonstrated in Figure 3).

## 4 Algorithms and Experiments

### 4.1 Sample-based Algorithms

Based on the analysis and results developed above, we provide sample-based estimates of the objective functions that lead to practical algorithms. Here we highlight several hyper-parameters: the learning rate $\alpha$; the trajectory truncation parameter (time horizon) $T$ (needs to be sufficiently large); the total sample size $N$ or the sampling interval $\delta_t$, with $N\cdot\delta_t=T$. Also denote $t_i := i\cdot\delta_t, i=0,\ldots,N-1$, for the time points that we observe data from the environment.

---

**Algorithm 1** CPG: Policy Gradient with $\exp(\beta)$ random rollout

**Input**: Policy parameters $\theta_0$, critic net parameters $\phi_0$, batch/sample size $J$

1: **for** $k=0,1,2,\cdots$ until $\theta_k$ converges **do**
2:     Collect a truncated trajectory $\{X_{t_i},a_{t_i},r_{t_i},p_{t_i}\}, i=1,\ldots,N$ from the environment using $\pi_{\theta_k}$.
3:     **for** $i=0,\ldots,N-1$ **do**: Update the critic parameters as in (8)
4:     **for** $j=1,,\ldots,J$ **do**: Draw i.i.d. $\tau_j$ from $\exp(\beta)$, round $\tau_j$ to the largest multiple of $\delta_t$ no larger than it, and compute the GAE estimator of $q(X_{\tau_j},a_{\tau_j})$

$$\tilde{q}(X_{\tau_j},a_{\tau_j}) := \left(r_{\tau_j}\delta_t + e^{-\beta\delta_t}V(X_{\tau_j+\delta_t})-V(X_{\tau_j})\right)/\delta_t \tag{25}$$

5:     Get an estimator of $\nabla_j\eta(\theta_k)$ as

$$\frac{1}{\beta}\left[\nabla_\theta\log\left(\pi^{\theta_k}(a_{\tau_j}\mid X_{\tau_j})\right)\left(\tilde{q}(X_{\tau_j},a_{\tau_j})+\gamma p(X_{\tau_j},a_{\tau_j},\pi^{\theta_k})\right)+\gamma\nabla_\theta p(X_{\tau_j},a_{\tau_j},\pi^{\theta_k})\right] \tag{26}$$

6:     Let $\tilde{\nabla}\eta(\theta_k)=\frac{1}{J}\sum_{j=1}^J\nabla_j\eta(\theta_k)$ and perform PG update: $\theta_{k+1}=\theta_k+\alpha\tilde{\nabla}\eta(\theta_k)$

---

**Continuous Policy Gradient (CPG).** To estimate the policy gradient (19) from data, we first sample an independent exponential variable $\tau \sim \exp(\beta)$ to get $(X_\tau^\pi, a_\tau^\pi) \sim (d_\mu^{\theta_0}, \pi^{\theta_0}(\cdot|x))$. If there is a $q$-function oracle, then we can obtain an unbiased estimate of the policy gradient (of which the convergence analysis follows [63]). Lack of such an oracle, we employ the generalized advantage estimation (GAE) technique [46] to get $q(X_t, a_t) \approx (Q_{\Delta t}(X_t, a_t; \pi) - V(X_t; \pi))/\delta_t \approx \left(r_t\delta_t + e^{-\beta\delta_t}V(X_{t+\delta_t}) - V(X_t)\right)/\delta_t$. This yields the policy gradient Algorithm 1.

**Continuous PPO (CPPO).** We now present Algorithm 2, a continuous version of the PPO, also as an approximation to the MM algorithm in Section 3.3. To do so, we need more hyper-parameters: the tolerance level $\epsilon$, and the KL-divergence radius $\delta$. Moreover, we set $\bar{D}_{\mathrm{KL}}(\theta\|\theta_k) := \mathbb{E}_{x \sim d_\mu^{\theta_k}} \sqrt{D_{\mathrm{KL}}(\pi_{\theta_k}(\cdot|x)\|\pi_\theta(\cdot|x))}$. (Empirically we find that taking average, instead of supremum, over $x$ does not affect the algorithm performance while reducing computational burden, similar to what's observed in the discrete-time TRPO in [45].)

---

**Algorithm 2** CPPO: PPO with adaptive penalty constant

---

**Input**: Policy parameters $\theta_0$, critic net parameters $\phi_0$
1: **for** $k = 0, 1, 2, \cdots$ until $\theta_k$ converge **do**
2:       Follow the same as Steps 2-6 in Algorithm 1.
3:       Compute policy update (by taking a fixed $s$ steps of gradient descent)

$$\theta_{k+1} = \arg\max_\theta \left\{ L^{\theta_k}(\theta) - C_{\mathrm{penalty}}^k \bar{D}_{\mathrm{KL}}(\theta\|\theta_k) \right\} \tag{27}$$

4:       **if** $\bar{D}_{\mathrm{KL}}(\theta_{k+1}\|\theta_k) \geq (1 + \epsilon)\delta$ **then**     $C_{\mathrm{penalty}}^{k+1} = 2C_{\mathrm{penalty}}^k$
5:       **else if** $\bar{D}_{\mathrm{KL}}(\theta_{k+1}\|\theta_k) \leq \delta/(1 + \epsilon)$ **then**     $C_{\mathrm{penalty}}^{k+1} = C_{\mathrm{penalty}}^k/2$

---

Algorithm 2 is essentially a continuous analogue of the TRPO/PPO methods. Note that in the penalty term we use the mean square-root of the KL-divergence, since we choose the radius $\delta < 1$; hence, the square-root distance will dominate in the bound in (23). Moreover, interestingly, throughout our primary experiments, using the square-root KL-divergence outperforms (using the KL-divergence itself). Refer to Appendix C,D for more details.

## 4.2 Experiments

**LQ stochastic control.** Consider an environment driven by an SDE with linear state dynamics and quadratic rewards, with $b(x, a) = Ax + Ba$, $\sigma(x, a) = Cx + Da$, where $A, B, C, D \in \mathbb{R}$, $p(x, a, \pi) = -\log(\pi(a|x))$, and $r(x, a) = -\left(\frac{M}{2}x^2 + Rxa + \frac{N}{2}a^2 + Px + Qa\right)$, where $M \geq 0, N > 0, R, P, Q \in \mathbb{R}$. Linear-quadratic (LQ) control problems play an important role in the control literature, not only because it has elegant and simple solutions but also because more complex, nonlinear problems can be approximated by LQ problems. In general, we do not know the model parameters (e.g., $A, B, \ldots$), and the idea is to use continuous RL methods to find the optimal policy.

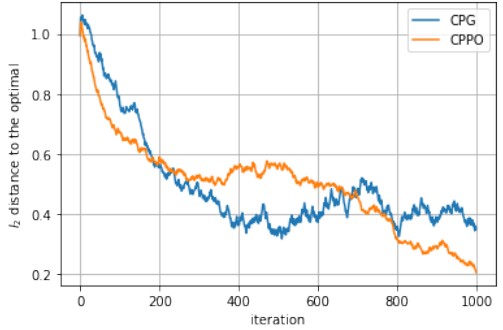
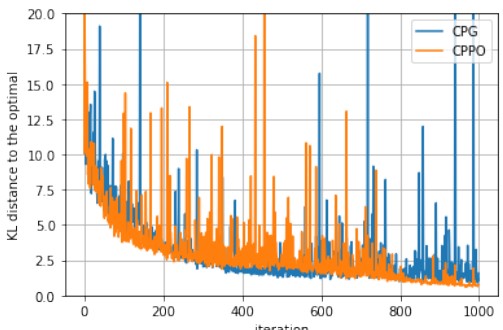

Figure 1: Convergence of $\theta$ in $l_2$ distance          Figure 2: Convergence of $\pi_\theta$ in KL-divergence

Here we adopt a Gaussian exploration parameterized by $\theta$ as: $\pi_\theta(\cdot \mid x) = \mathcal{N}(\theta_1 x + \theta_2, \exp(\theta_3))$, and we also parameterize the value function by $\phi$ as $V_\phi(x) = \frac{1}{2}\phi_2 x^2 + \phi_1 x + \phi_0$. (In fact, as shown in [58, Theorem 4], the optimal exploration and value functions are of this form, and constants such as $\theta$ and $\phi$ can be computed explicitly given the model dynamics.) We randomly choose a set of initial constants, and compute the optimal $\theta^*$ and $\phi^*$ with respect to these parameters; refer to Appendix D.1 for more details. Figure 1,2 show the convergence of algorithms for one certain realized trajectory.

In Figure 1, we compute the $l_2$ distance between the current policy parameters and the optimal ones, i.e. $\|\theta_k - \theta^*\|_2$, which tracks the convergence of the policy parameters. In Figure 2, we plot the sample estimated KL divergence between the current policy $\pi_k$ (specified by $\theta_k$) and $\pi^*$ (specified by $\theta^*$), i.e. $\mathbb{E}_{x \sim d_\mu^{\theta_k}} D_{\mathrm{KL}}(\pi_{\theta_k}(\cdot|x)\|\pi_\theta(\cdot|x))$. The reason to consider the KL-divergence between $\pi_k$ and $\pi^*$ is that minimizing the KL-divergence to the optimal solution is equivalent to minimizing the distance between the current policy objective and the optimal objective (see Appendix D.1)). The experiments illustrate that our proposed algorithms do converge to the (local) optimum.

We also compare the performance of CPO and CPPO to the approaches that directly discretize the time, and then apply the classical discrete-time PG and PPO algorithms. See the details in Appendix D.4. The experiments show that our proposed CPO and CPPO are comparable in terms of sample efficiency, and in many cases they outperform the discrete-time algorithms under a range of time discretization.

**2-dimensional optimal pair trading**. We also consider the 2-dimensional optimal pair trading problem formulated in [36]. The state space is $X = (S, W) \in \mathbb{R}^2$ with $X(0) = (s_0, w_0)$, where $S$ represents the spread between two stocks, and $W$ denotes the corresponding wealth process. The trader intends to maximize the total discounted reward, with the reward function $r(X, a) = \log(1 + W)$. The state dynamics are:

$$\mathrm{d}S_t = k(\theta - S_t)\mathrm{d}t + \eta\mathrm{d}B_t, \quad \mathrm{d}W_t = a_t W_t(k(\theta - S_t) + \frac{1}{2}\eta^2 + \rho\sigma\eta + r_f)\mathrm{d}t + \eta W_t\mathrm{d}B_t, \quad (28)$$

We set $p(x, a, \pi) \equiv 0$, and add a constraint on the action: $a_t \in [-\ell, \ell]$. (The action $a_t$ is the position taken on the first stock, which can be long/positive or short/negative.) Since the action space is bounded and continuous, we consider a beta distribution for policy parameterization: $\pi_\theta(a \mid X) := f\left(\frac{a+\ell}{2\ell}, \alpha_\theta(X), \beta_\theta(X)\right)$ with $f(x, \alpha, \beta) := \frac{\Gamma(\alpha+\beta)}{\Gamma(\alpha)\Gamma(\beta)}x^{\alpha-1}\left(1 - x^{\beta-1}\right)$. For $\alpha_\theta$ and $\beta_\theta$, we use a 3-layer neural network (NN) parameterized by $\theta$ for function approximation; and use another 3-layer NN for value function approximation. (More details are provided in Appendix D.2.)

Figure 3 shows that both algorithms, CPG and CPPO, converge to a local optimum (different between the two), and with an overall increasing trend over iterations. (Averaging is taken over 100 Monte Carlo estimates for each policy evaluation.)

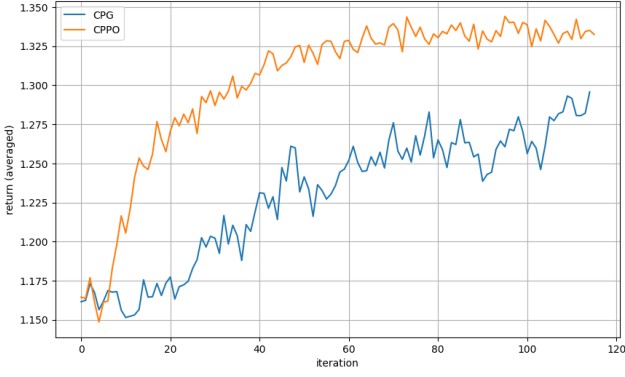

Figure 3: Performance of both algorithms to the task

# 5 Conclusion and Further Works

We have developed in this paper the basic theoretical framework for policy optimization in continuous RL, and illustrated its potential applications using numerical experiments.

For further research, two topics are high on our agenda. First, we plan to study the convergence (rate) of the continuous policy gradient and TRPO/PPO, vis-a-vis the error due to the time increment $\delta_t$. Our conjecture is that it is likely to be polynomial-bounded under mild assumptions, similar to the analysis in [20, 16]), thus extending beyond the condition required by [63] and [60, 31]. Second, for the bounds on the statistical distance and the performance difference, we want to further develop a consistent bound like the one in [11] (for the discrete setting), i.e., one that remains meaningful when the discount factor $\beta \to 0$.

## Acknowledgments

Wenpin Tang gratefully acknowledges financial support through NSF grants DMS-2113779 and DMS-2206038, and through a start-up grant at Columbia University. The works of Hanyang Zhao and David Yao are part of a Columbia-CityU/HK collaborative project that is supported by InnotHK Initiative, The Government of the HKSAR and the AIFT Lab. We also thank Yanwei Jia and Xun Yu Zhou for helpful discussions.

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

# Appendix A    Continuous RL: Formulation and Well-Posedness

## A.1    Exploratory Stochastic-Control

For $n, m$ positive integers, let $b : \mathbb{R}^n \times \mathcal{A} \mapsto \mathbb{R}^n$ and $\sigma : \mathbb{R}^n \times \mathcal{A} \mapsto \mathbb{R}^{n \times m}$ be given functions, where $\mathcal{A}$ is a compact action space. A classical stochastic control problem [15, 62] is to control the state (or feature) dynamics governed by an Itô process, defined on a filtered probability space $\left( \Omega, \mathcal{F}, \mathbb{P}; \{ \mathcal{F}_s^B \}_{s \geq 0} \right)$, along with an $\{ \mathcal{F}_s^B \}$-Brownian motion $B = \{ B_s, s \geq 0 \}$:

$$\mathrm{d}X_s^a = b \left( X_s^a, a_s \right) \mathrm{d}s + \sigma \left( X_s^a, a_s \right) \mathrm{d}B_s, \ s \geq t, \quad X_t = x, \tag{29}$$

where $a_s$ is the agent's action (control) at time $s$. The goal of the stochastic control (discounted objective over an infinite time horizon) is for any time-state pair $(t, x)$ in (29), to find the optimal $\{ \mathcal{F}_s^B \}_{s \geq 0}$-progressively measurable sequence of actions $a = \{ a_s, s \geq t \}$ (called the optimal policy) that maximizes the expected total $\beta$-discounted reward:

$$\mathbb{E} \left[ \int_t^{+\infty} e^{-\beta(s-t)} r \left( X_s^a, a_s \right) \mathrm{d}s \mid X_t^a = x \right], \tag{30}$$

where $r : \mathbb{R}^n \times \mathcal{A} \mapsto \mathbb{R}$ is the running reward of the current state and action $(X_s^a, a_s)$, and $\beta > 0$ is a discount factor that measures the time-depreciation of the objective value (or the impatience level of the agent). Note that the state process $X^a = \{ X_s^a, s \geq t \}$ depends on the starting (initial) time-state pair $(t, x)$. For ease of notation, we denote by $X^a$ instead of $X^{t,x,a} = \{ X_s^{t,x,a}, s \geq t \}$ the solution to the SDE in (29) when there is no ambiguity.

Listed below are the standard assumptions to ensure the well-posedness of the stochastic control problem in (29)-(30).

**Assumption 2.** *The following conditions are assumed throughout:*
*(i) $b, \sigma, r$ are all continuous functions in their respective arguments;*
*(ii) $b, \sigma$ are uniformly Lipschitz continuous in $x$, i.e., there exists a constant $C > 0$ such that for $\varphi \in \{ b, \sigma \}$,*

$$\left\| \varphi(x, a) - \varphi \left( x', a \right) \right\|_2 \leq C \left\| x - x' \right\|_2, \quad \text{for all } a \in \mathcal{A}, \ x, x' \in \mathbb{R}^n; \tag{31}$$

*(iii) $b, \sigma$ have linear growth in $x$ and $a$, i.e., there exists a constant $C > 0$ such that for $\varphi \in \{ b, \sigma \}$,*

$$\| \varphi(x, a) \|_2 \leq C(1 + \| x \|_2 + \| a \|_2), \quad \text{for all } (x, a) \in \mathbb{R}^n \times \mathcal{A}; \tag{32}$$

*(iv) $r$ has polynomial growth in $x$ and $a$, i.e., there exists a constant $C > 0$ and $\mu \geq 1$ such that*

$$|r(x, a)| \leq C \left( 1 + \| x \|_2^\mu + \| a \|_2^\mu \right) \quad \text{for all } (x, a) \in \mathbb{R}^n \times \mathcal{A}. \tag{33}$$

The key idea underlying *exploratory* stochastic control is to use a randomized policy (or relaxed control), i.e., apply a probability distribution to the admissible action space. To do so, let's assume the probability space is rich enough to support a uniform random variable $Z$ that is independent of the Brownian motion $B = \{ B_t \}$. We then expand the original filtered probability space to $\left( \Omega, \mathcal{F}, \mathbb{P}; \{ \mathcal{F}_s \}_{s \geq 0} \right)$, where $\mathcal{F}_s = \mathcal{F}_s^B \vee \sigma(Z)$ (i.e., augment $\mathcal{F}_s^B$ with the sigma field generated by $Z$).

Let $\pi : \mathbb{R}^n \ni x \mapsto \pi(\cdot \mid x) \in \mathcal{P}(\mathcal{A})$ be a stationary feedback policy given the state at $x$, where $\mathcal{P}(\mathcal{A})$ is a suitable collection of probability distributions (with density functions). At each time $s$, an action $a_s$ is generated from the distribution $\pi \left( \cdot \mid X_s^a \right)$, i.e. the policy only depends on the current state. In other words, we only consider stationary, or time-independent feedback control policies for the stochastic control problem (29)-(30).

Given a stationary policy $\pi \in \mathcal{P}(\mathcal{A})$, an initial state $x$, and an $\{ \mathcal{F}_s \}$-progressively measurable action process $a^\pi = \{ a_s^\pi, s \geq 0 \}$ generated from $\pi$, the state process $X^\pi = \{ X_s^\pi, s \geq 0 \}$ follows:

$$\mathrm{d}X_s^\pi = b \left( X_s^\pi, a_s^\pi \right) \mathrm{d}s + \sigma \left( X_s^\pi, a_s^\pi \right) \mathrm{d}B_s, \ s \geq t, \quad X_0^\pi = x, \tag{34}$$

defined on $\left( \Omega, \mathcal{F}, \mathbb{P}; \{ \mathcal{F}_s \}_{s \geq 0} \right)$. It is easy to see that the dynamics in (34) define a time-homogeneous Markov process, such that for each $t \geq 0$ and $x$:

$$\left( X_s^\pi \mid X_0^\pi = x \right) \overset{d}{=} \left( X_{s+t}^\pi \mid X_t^\pi = x \right), \ s \geq 0.$$

Consequently, the objective in (30) is independent of time $t$, and is equal to:

$$\mathbb{E}\left[\int_0^{+\infty} e^{-\beta s} r\left(X_s^\pi, a_s^\pi\right) \mathrm{d}s \mid X_0^\pi = x\right]. \tag{35}$$

Furthermore, following [58], we can add a regularizer to the objective function to encourage exploration (represented by the randomized policy), leading to

$$V(t, x; \pi) := \mathbb{E}\left[\int_t^\infty e^{-\beta(s-t)}\left[r\left(X_s^\pi, a_s^\pi\right) + \gamma p\left(X_s^\pi, a_s^\pi, \pi\left(\cdot \mid X_s^\pi\right),\right)\right] \mathrm{d}s \mid X_t^\pi = x\right], \tag{36}$$

where $p : \mathbb{R}^n \times \mathcal{A} \times \mathcal{P}(\mathcal{A}) \mapsto \mathbb{R}$ is the regularizer, and $\gamma \geq 0$ is a weight parameter on exploration (also known as the "temperature" parameter). For instance, in [58], $p$ is taken as the differential entropy,

$$p(x, a, \pi(\cdot)) := -\log \pi(a),$$

and hence, the "entropy" regularizer. The same argument as before justifies that $V(t, x; \pi)$ is independent of time $t$. That is, for all $t \geq 0$,

$$V(t, x; \pi) \equiv V(x; \pi) := \mathbb{E}^\mathbb{P}\left[\int_0^\infty e^{-\beta s}\left[r\left(X_s^\pi, a_s^\pi\right) + \gamma p\left(X_s^\pi, a_s^\pi, \pi\left(\cdot \mid X_s^\pi\right)\right)\right] \mathrm{d}s \mid X_0^\pi = x\right]; \tag{37}$$

which is the state-value function under the policy $\pi$, $V(x; \pi)$, in (4), and which, in turn, leads to the performance function $\eta(\pi)$ in (6). Moreover, recall the main task of the continuous RL is to find (or approximate) $\eta^* = \max_\pi \eta(\pi)$, where $\max$ is over all admissible policies.

## A.2 Controlled SDE and the HJ Equation

Note that the exploratory state dynamics in (34) is governed by a general Itô process. It is sometimes more convenient to consider an equivalent SDE representation— in the sense that its (weak) solution has the same distribution as the Itô process in (34) at each fixed time $t$. It is known ([58]) that when $n = m = 1$, the marginal distribution of $\{X_s^\pi, s \geq 0\}$ agrees with that of the solution to the SDE, denoted by $\{\tilde{X}_s, s \geq 0\}$:

$$\mathrm{d}\tilde{X}_s = \tilde{b}\left(\tilde{X}_s, \pi\left(\cdot \mid \tilde{X}_s\right)\right)\mathrm{d}s + \tilde{\sigma}\left(\tilde{X}_s, \pi\left(\cdot \mid \tilde{X}_s\right)\right)\mathrm{d}\tilde{B}_s, \quad \tilde{X}_0 = x,$$

where $\tilde{b}(x, \pi(\cdot)) = \int_\mathcal{A} b(x, a)\pi(a)\mathrm{d}a$ and $\tilde{\sigma}(x, \pi(\cdot)) = \sqrt{\int_\mathcal{A} \sigma^2(x, a)\pi(a)\mathrm{d}a}$. This result is easily extended to arbitrary $n, m$, thanks to [7, Corollary 3.7], with the precise statement presented below (assuming $n = m$ for ease of exposition).

**Theorem 6.** *Assume that for a policy $\pi$ and for every $x$,*

$$\int_\mathcal{A} \sigma^2(x, a)\pi(a)\mathrm{d}a \in \mathbb{R}^{n \times n},$$

*is positive definite. Then there exists a filtered probability space $\left(\tilde{\Omega}, \tilde{\mathcal{F}}, \left\{\tilde{\mathcal{F}}_t\right\}_{t \geq 0}, \tilde{\mathbb{P}}\right)$ that supports a continuous $\mathbb{R}^n$-valued adapted process $\tilde{X}$ and an $n$-dimensional Brownian motion $\tilde{B}$ satisfying*

$$\mathrm{d}\tilde{X}_s = \tilde{b}\left(\tilde{X}_s, \pi\left(\cdot \mid \tilde{X}_s\right)\right)\mathrm{d}s + \tilde{\sigma}\left(\tilde{X}_s, \pi\left(\cdot \mid \tilde{X}_s\right)\right)\mathrm{d}\tilde{B}_s, \quad \tilde{X}_0 = x, \tag{38}$$

*where*

$$\tilde{b}(x, \pi(\cdot)) = \int_\mathcal{A} b(x, a)\pi(a)\mathrm{d}a, \quad \tilde{\sigma}(x, \pi(\cdot)) = \left(\int_\mathcal{A} \sigma^2(x, a)\pi(a)\mathrm{d}a\right)^{\frac{1}{2}}.$$

*For each $s \geq 0$, the distribution of $\tilde{X}_s$ under $\tilde{\mathbb{P}}$ agrees with that of $X_s^\pi$ under $\mathbb{P}$ defined in (34).*

As a consequence, the state value function in (37) is identical to

$$V(x; \pi) = \mathbb{E}\left[\int_0^\infty e^{-\beta s}\int_\mathcal{A}\left[r(\tilde{X}_s, a) + \gamma p\left(\tilde{X}_s, a, \pi(\cdot \mid \tilde{X}_s)\right)\right]\pi(a \mid \tilde{X}_s)\mathrm{d}a\,\mathrm{d}s \mid \tilde{X}_0 = x\right].$$

Also define

$$\tilde{r}(x,\pi) = \int_{\mathcal{A}} r(x,a)\pi(a|s)\mathrm{d}a, \quad \tilde{p}(x,\pi) = \int_{\mathcal{A}} p(x,a,\pi)\pi(a|x)\mathrm{d}a,$$

so we can simplify the value function to

$$V(x;\pi) = \mathbb{E}\left[\int_0^\infty e^{-\beta s}\left[\tilde{r}(\tilde{X}_s,\pi) + \gamma\tilde{p}\left(\tilde{X}_s^\pi,\pi(\cdot\mid\tilde{X}_s)\right)\right]\,\mathrm{d}s\mid\tilde{X}_0 = x\right]. \tag{39}$$

Following the principle of optimality, $V$ then satisfies the HJ equation:

$$\beta V(x;\pi) - \tilde{b}(x,\pi)\cdot\nabla V(x;\pi) - \frac{1}{2}\tilde{\sigma}^2(x,\pi)\circ\nabla^2 V(x;\pi) - \tilde{r}(x,\pi) - \gamma\tilde{p}(x,\pi) = 0. \tag{40}$$

To guarantee that the HJ equation in (40) characterizes the state-value function in (39), we need

**Assumption 3.** *Assume the following conditions hold:*
*(i) $b,\sigma,r,p$ are all continuous functions in their respective arguments.*
*(ii) $b,r,p$ are uniformly Lipschitz continuous in $x$, i.e., there exists a constant $C > 0$ such that for $\varphi \in \{b,r\}$,*

$$\|\varphi(x,a) - \varphi(x',a)\|_2 \le C\|x - x'\|_2, \quad \text{for all } a \in \mathcal{A},\ x,x' \in \mathbb{R}^n,$$

*and*

$$|p(x,a,\pi) - p(x',a,\pi)| \le C\|x - x'\|_2, \quad \text{for all } a \in \mathcal{A},\ \pi \in \mathcal{P}(\mathcal{A}),\ x,x' \in \mathbb{R}^n.$$

*(iii) $\tilde{\sigma}$ is globally bounded, i.e., there exist $0 < \sigma_0 < \bar{\sigma}_0$ such that*

$$\sigma_0^2\cdot I \le \tilde{\sigma}^2(x,a) \le \bar{\sigma}_0^2\cdot I, \quad \text{for all } a \in \mathcal{A},\ x \in \mathbb{R}^n.$$

*(iv) the SDE (38) has a weak solution which is unique in distribution.*
*(v) $\pi(a|x)$ is measurable in $(x,a)$ and is uniformly Lipschitz continuous in $x$, i.e., there exists a constant $C > 0$ such that*

$$\int_{\mathcal{A}} |\pi(a|x) - \pi(a|x')|\,da \le C\|x - x'\|_2, \quad \text{for all } x,x' \in \mathbb{R}^n.$$

**Theorem 7.** *Under Assumption 3, the state-value function in (39) is the unique (subquadratic) viscosity solution to the HJ equation in (40).*

*Proof.* By [56, Section 3.1], the HJ equation in (40) has a unique (subquadratic) viscosity solution under the conditions (i)-(iii). Further by [21, Lemma 2], the viscosity solution is the state-value function. $\qquad\square$

## Appendix B    Proofs of Main Results (in §3)

### B.1    Proof of Theorem 2

Recall in the proof sketch of the Theorem in §3, we have defined the operator $\mathcal{L}^\pi : C^2(\mathbb{R}^n) \mapsto C(\mathbb{R}^n)$ as

$$(\mathcal{L}^\pi\varphi)(x) := -\beta\varphi(x) + \tilde{b}(x,\pi)\cdot\nabla\varphi(x) + \frac{1}{2}\tilde{\sigma}(x,\pi)^2\circ\nabla^2\varphi(x),$$

which leads to the following characterization of the HJ equation:

$$-\mathcal{L}^\pi V(x;\pi) = \tilde{r}(x,\pi) + \gamma\tilde{p}(x,\pi). \tag{41}$$

We need the following two lemmas concerning the operator $\mathcal{L}^\pi$.

**Lemma 8.** *For any $\varphi \in C^2(\mathbb{R}^n)$, we have*

$$\int_{\mathbb{R}^n} d_x^\pi(y)(-\mathcal{L}^\pi\varphi)(y)\mathrm{d}y = \varphi(x).$$

*Proof.* The left hand side of the above equation is

$$
= \quad \mathbb{E} \int_0^\infty e^{-\beta s} \left( \beta \varphi(\tilde{X}_s^\pi) - \tilde{b}(\tilde{X}_s^\pi, \pi) \frac{\partial \varphi}{\partial x}(\tilde{X}_s^\pi) - \frac{1}{2} \tilde{\sigma}(\tilde{X}_s^\pi, \pi)^2 \frac{\partial^2 \varphi}{\partial x^2}(\tilde{X}_s^\pi) \right) \mathrm{d}s
$$

$$
= \quad \mathbb{E} \int_0^\infty e^{-\beta s} \left[ \left( \beta \varphi(\tilde{X}_s^\pi) - \tilde{b}(\tilde{X}_s^\pi, \pi) \frac{\partial \varphi}{\partial x}(\tilde{X}_s^\pi) - \frac{1}{2} \tilde{\sigma}(\tilde{X}_s^\pi, \pi)^2 \frac{\partial^2 \varphi}{\partial x^2}(\tilde{X}_s^\pi) \right) \mathrm{d}s - \tilde{\sigma}(\tilde{X}_s^\pi, \pi) \frac{\partial \varphi}{\partial x}(\tilde{X}_s^\pi) \mathrm{d}B_s \right]
$$

$$
= \quad \mathbb{E} \int_0^\infty \mathrm{d} \left( -e^{-\beta s} \varphi(\tilde{X}_s^\pi) \right)
$$

$$
= \quad \lim_{s \to \infty} \left( -e^{-\beta s} \varphi(\tilde{X}_s^\pi) \right) + \varphi(\tilde{X}_0^\pi)
$$

$$
= \quad \varphi(x),
$$

where the first equality follows from the definition of the occupation time and the third equality from Itô's formula. $\qquad\square$

**Lemma 9.** *Let $\pi, \hat{\pi}$ be two feedback policies. We have*

$$
(\mathcal{L}^{\hat{\pi}} - \mathcal{L}^\pi) V(x; \pi) + \tilde{r}(x, \hat{\pi}) - \tilde{r}(x, \pi) - \gamma \tilde{p}(x, \pi) = \int_{\mathcal{A}(x)} \hat{\pi}(a \mid x) q(x, a; \pi) \mathrm{d}a. \qquad (42)
$$

*Proof.* By definition of $q(x, a; \pi)$ in (11), we have

$$
\text{RHS} \quad = \quad \int_{\mathcal{A}(x)} \hat{\pi}(a \mid x) \left( \mathcal{H}^a \left( x, \frac{\partial V}{\partial x}(x; \pi), \frac{\partial^2 V}{\partial x^2}(x; \pi) \right) - \beta V(x; \pi) \right) \mathrm{d}a
$$

$$
= \quad \int_{\mathcal{A}(x)} \hat{\pi}(a \mid x) \left( b(x, a) \cdot \frac{\partial V}{\partial x}(x; \pi) + \frac{1}{2} \sigma^2(x, a) \circ \frac{\partial^2 V}{\partial x^2}(x; \pi) + r(x, a) - \beta V(x; \pi) \right) \mathrm{d}a
$$

$$
= \quad \tilde{r}(x, \hat{\pi}) + \mathcal{L}^{\hat{\pi}} V^\pi(x)
$$

$$
= \quad \tilde{r}(x, \hat{\pi}) - \tilde{r}(x, \pi) - \gamma \tilde{p}(x, \pi) + \mathcal{L}^{\hat{\pi}} V^\pi(x) - \mathcal{L}^\pi V^\pi(x)
$$

$$
= \quad \text{LHS}.
$$

$\qquad\square$

*Proof of Theorem 2.* Note that in (13), the equation to be proven, the right hand side can be written as $\int_{\mathbb{R}} d_\mu^{\hat{\pi}}(y) f(x; \pi, \hat{\pi}) \mathrm{d}y$, with

$$
f(x; \pi, \hat{\pi}) := \int_{\mathcal{A}} \hat{\pi}(a \mid x) \left( q(x, a; \pi) + \gamma p(x, a, \hat{\pi}) \right) \mathrm{d}a.
$$

From Lemma 9, we have

$$
f(x; \pi, \hat{\pi}) = (\mathcal{L}^{\hat{\pi}} - \mathcal{L}^\pi) V(x; \pi) + \tilde{r}(x, \hat{\pi}) + \gamma \tilde{p}(x, \hat{\pi}) - \tilde{r}(x, \pi) - \gamma \tilde{p}(x, \pi). \qquad (43)
$$

On the other hand, for the left hand side of (13), we have

$$
\eta(\pi) = \int_{\mathbb{R}^n} V(y; \pi) \mu(\mathrm{d}y) = \int_{\mathbb{R}^n} d_\mu^{\hat{\pi}}(y) (-\mathcal{L}^{\hat{\pi}}) V(y; \pi) \mathrm{d}y, \qquad (44)
$$

with the second equality following from Lemma 8; and

$$
\eta(\hat{\pi}) = \int_{\mathbb{R}} d_\mu^{\hat{\pi}}(y) \left[ \tilde{r}(y, \hat{\pi}) + \gamma \tilde{p}(y, \hat{\pi}) \right] \mathrm{d}y, \qquad (45)
$$

following the definition of the discounted expected occupation time; moreover, from (41), we have

$$
0 = \int_{\mathbb{R}} d_\mu^{\hat{\pi}}(y) \left[ (-\mathcal{L}^\pi) V(y; \pi) - \tilde{r}(y, \pi) - \gamma \tilde{p}(y, \pi) \right] \mathrm{d}y. \qquad (46)
$$

Hence, combining the last three equations (44,45,46), we have

$$
\eta(\hat{\pi}) - \eta(\pi) = \int_{\mathbb{R}} d_\mu^{\hat{\pi}}(y) \left[ (\mathcal{L}^{\hat{\pi}} - \mathcal{L}^\pi) V(y; \pi) + \tilde{r}(y, \hat{\pi}) + \gamma \tilde{p}(y, \hat{\pi}) - \tilde{r}(y, \pi) - \gamma \tilde{p}(y, \pi) \right] \mathrm{d}y. \qquad (47)
$$

Thus, we have shown LHS=RHS in (13). $\qquad\square$

## B.2 Proof of Theorem 3

*Proof.* It suffices to show the integral version of the theorem:

$$\nabla_\theta \left( \eta(\pi^\theta) \right) \big|_{\theta=\theta} = \int_{\mathbb{R}^n} d_\mu^{\pi^\theta}(x) \left[ \int_{\mathcal{A}} \nabla_\theta \pi^\theta(a \mid x) \left( q(x, a; \pi^\theta) + \gamma p(x, a, \pi^\theta) \right) + \right.$$

$$\left. \gamma \cdot \pi^\theta(a \mid x) \nabla_\theta p(x, a, \pi^\theta) \mathrm{d}a \right] \mathrm{d}x. \tag{48}$$

As before, we simplify notation by denoting $\eta(\pi^\theta)$ as $\eta(\theta)$ and $d^{\pi^\theta}$ as $d^\theta$. Then, by Theorem 2), we have

$$\eta(\theta + \delta\theta) - \eta(\theta) = \int_{\mathbb{R}^n} d_\mu^{\theta+\delta\theta}(x) \left[ \int_{\mathcal{A}} \pi^{\theta+\delta\theta}(a \mid x) \left( q(x, a; \theta) + \gamma p(x, a, \theta + \delta\theta) \right) \mathrm{d}a \right] \mathrm{d}x. \tag{49}$$

Denote

$$f(\delta\theta) = \int_{\mathcal{A}} \pi^{\theta+\delta\theta}(a \mid x) \left( q(x, a; \theta) + \gamma p(x, a, \theta + \delta\theta) \right) \mathrm{d}a.$$

Note that $f(0) = 0$, which follows from

$$f(0) = \int_{\mathcal{A}} \pi^\theta(a \mid x) \left( q(x, a; \theta) + \gamma p(x, a, \theta) \right) \mathrm{d}a$$

$$= \int_{\mathcal{A}} \pi^\theta(a \mid x) \left( \mathcal{H}^a(x, \frac{\partial V}{\partial x}(x; \pi), \frac{\partial^2 V}{\partial x^2}(x; \pi)) - \beta V(x; \pi) + \gamma p(x, a, \theta) \right) \mathrm{d}a$$

$$= -\beta V(x; \pi) + \tilde{b}(x, \pi) \cdot \nabla V(x; \pi) + \frac{1}{2}\tilde{\sigma}^2(x, \pi) \circ \nabla^2 V(x; \pi) + \tilde{r}(x, \pi) + \gamma \tilde{p}(x, \pi)$$

$$= 0.$$

Thus,

$$\eta(\theta + \delta\theta) - \eta(\theta) = \langle d_\mu^{\theta+\delta\theta}, f(\delta\theta) \rangle$$

$$= \langle d_\mu^{\theta+\delta\theta}, f(\delta\theta) \rangle - \langle d_\mu^{\theta+\delta\theta}, f(0) \rangle$$

$$= \langle d_\mu^{\theta+\delta\theta}, f(\delta\theta) - f(0) \rangle$$

$$= \langle d_\mu^{\theta+\delta\theta} - d_\mu^\theta, f(\delta\theta) - f(0) \rangle + \langle d_\mu^\theta, f(\delta\theta) - f(0) \rangle.$$

Dividing both sides by $\delta\theta$ completes the proof, as the first term on the last line above is of higher order than $\delta\theta$. $\qquad\square$

## B.3 Proofs of Lemma 4 and Theorem 5

We need a lemma for the perturbation bounds.

**Lemma 10.** *Assume that both $\tilde{\sigma}^2(x, \hat{\pi}(\cdot))$ and $\tilde{\sigma}^2(x, \pi(\cdot))$ are positive definite and*

$$\tilde{\sigma}^2(x, \pi(\cdot)), \tilde{\sigma}^2(x, \hat{\pi}(\cdot)) \geq \sigma_0^2 \cdot I.$$

*where $\sigma_0 > 0$, then we have that the difference between the square root matrix is bounded by*

$$\|\tilde{\sigma}(x, \hat{\pi}) - \tilde{\sigma}(x, \pi)\|_2 \leq \frac{1}{2\sigma_0} \|\tilde{\sigma}^2(x, \hat{\pi}) - \tilde{\sigma}^2(x, \pi)\|_2.$$

*If we also assume that the upper bounds, i.e.*

$$\tilde{\sigma}^2(x, \pi(\cdot)), \tilde{\sigma}^2(x, \hat{\pi}(\cdot)) \leq \bar{\sigma}_0^2 \cdot I.$$

*by some $\bar{\sigma}_0 > \sigma_0 > 0$, then we have*

$$\|\tilde{\sigma}(x, \hat{\pi}) - \tilde{\sigma}(x, \pi)\|_2 \leq \frac{\bar{\sigma}_0}{2\sigma_0} \|\hat{\pi} - \pi\|_1^{\frac{1}{2}}.$$

*Proof.* Consider a normalized vector $x$ with $\|x\|_2 = 1$ is an eigenvector of $A^{\frac{1}{2}} - B^{\frac{1}{2}}$ with eigenvalue $\mu$ then

$$x^T(A - B)x = x^T(A^{\frac{1}{2}} - B^{\frac{1}{2}})A^{\frac{1}{2}}x + x^T B^{\frac{1}{2}}(A^{\frac{1}{2}} - B^{\frac{1}{2}})x$$

$$= \mu x^T(A^{\frac{1}{2}} + B^{\frac{1}{2}})x.$$

thus, if $A, B \geq \sigma_0^2 I$, this implies

$$\mu \leq \frac{|x^T(A-B)x|}{x^T(A^{\frac{1}{2}}+B^{\frac{1}{2}})x} \leq \|A-B\|_2 \cdot \lambda_{\min}(A^{\frac{1}{2}}+B^{\frac{1}{2}})^{-1} \leq \|A-B\|_2/(2\sigma_0).$$

Furthermore, note that

$$\tilde{\sigma}^2(x,\hat{\pi}) - \tilde{\sigma}^2(x,\pi) = \int_{\mathcal{A}} \sigma^2(x,a)(\tilde{\pi}(a|x) - \pi(a|x))\mathrm{d}a.$$

so

$$\|\tilde{\sigma}^2(x,\hat{\pi}) - \tilde{\sigma}^2(x,\pi)\|_2 \leq \bar{\sigma}_0^2 \int_{\mathcal{A}} |\tilde{\pi}(a|x) - \pi(a|x)|\mathrm{d}a = \bar{\sigma}_0^2 \cdot \|\tilde{\pi}(a|x) - \pi(a|x)\|_1.$$

$\square$

*Proof* (of Lemma 4). Consider the Wasserstein-2 distance $W_2(\mu, v)$ between distribution $\mu$ and $v$ as

$$W_2(\mu, \nu) = \left( \inf_{\gamma \in \Gamma(\mu,\nu)} \mathbf{E}_{(x,y) \sim \gamma} \|x-y\|_2^2 \right)^{1/2},$$

where $\Gamma(\mu, \nu)$ is the set all probability measures on the product space $\mathbb{R}^n \times \mathbb{R}^n$ with the marginal distributions being $\mu$ and $v$, and $\|\cdot\|_2$ is the standard Euclidean distance. Denote

$$\bar{d}_\mu^\pi := \beta d_\mu^\pi.$$

We want to get an upper bound on $W_2(\bar{d}_\mu^\pi, \bar{d}_\mu^{\hat{\pi}})$ in terms of the distance between two policies $\pi$ and $\hat{\pi}$. Consider a specific coupling $(X_t, Y_t)$ below:

$$\begin{cases} \mathrm{d}X_s = \tilde{b}\left(X_s, \pi\left(\cdot \mid X_s\right)\right)\mathrm{d}s + \tilde{\sigma}\left(X_s, \pi\left(\cdot \mid X_s\right)\right)\mathrm{d}B_s, \\ \mathrm{d}Y_s = \tilde{b}\left(Y_s, \hat{\pi}\left(\cdot \mid Y_s\right)\right)\mathrm{d}s + \tilde{\sigma}\left(Y_s, \hat{\pi}\left(\cdot \mid Y_s\right)\right)\mathrm{d}B_s. \end{cases} \tag{50}$$

with $X_0 = Y_0$, which leads to a joint distribution over $\mathbb{R}^n \times \mathbb{R}^n$:

$$\tilde{\gamma} := \left\{ \tilde{p}(x,y) = \int_0^\infty \frac{1}{\beta} e^{-\beta t} f_{(X_t, Y_t)}(x,y)\mathrm{d}t \right\}.$$

Hence,

$$W_2^2(\bar{d}_\mu^\pi, \bar{d}_\mu^{\hat{\pi}}) \leq \mathbb{E}_{(x,y) \sim \tilde{\gamma}} \|x-y\|_2^2 = \int_0^\infty \frac{1}{\beta} e^{-\beta s} \mathbb{E}\|X_s - Y_s\|_2^2 \mathrm{d}s. \tag{51}$$

It then boils down to estimating $\mathbb{E}\|X_s - Y_s\|_2^2$. By Itô's formula,

$$\mathrm{d}\|X_s - Y_s\|_2^2 = 2(X_s - Y_s)^\top \left[ (\tilde{b}\left(X_s, \pi\right) - \tilde{b}\left(Y_s, \hat{\pi}\right))\mathrm{d}s + (\tilde{\sigma}\left(X_s, \pi\right) - \tilde{\sigma}\left(Y_s, \hat{\pi}\right))\mathrm{d}B_s \right]$$
$$+ \mathrm{Tr}\left[ (\tilde{\sigma}\left(X_s, \pi\right) - \tilde{\sigma}\left(Y_s, \hat{\pi}\right))^2 \right]\mathrm{d}s.$$

Taking expectation on both sides yields

$$\frac{\mathrm{d}}{\mathrm{d}s}\mathbb{E}\|X_s - Y_s\|_2^2 = \underbrace{2\mathbb{E}\left[ (X_s - Y_s)^\top (\tilde{b}\left(X_s, \pi\right) - \tilde{b}\left(Y_s, \hat{\pi}\right))\mathrm{d}s \right]}_{(A)} + \underbrace{\mathrm{Tr}\left[ \mathbb{E}(\tilde{\sigma}\left(X_s, \pi\right) - \tilde{\sigma}\left(Y_s, \hat{\pi}\right))^2 \right]}_{(B)},$$

$$\tag{52}$$

with

$$(A) = \mathbb{E}\left[ (X_s - Y_s)^\top (\tilde{b}\left(X_s, \pi\right) - \tilde{b}\left(Y_s, \pi\right))\mathrm{d}s \right] + \mathbb{E}\left[ (X_s - Y_s)^\top (\tilde{b}\left(Y_s, \pi\right) - \tilde{b}\left(Y_s, \hat{\pi}\right))\mathrm{d}s \right]$$

$$\leq C_{\tilde{b}} \cdot \mathbb{E}\|X_s - Y_s\|_2^2 + \frac{1}{2}\mathbb{E}\|X_s - Y_s\|_2^2 + \frac{1}{2}\mathbb{E}\|\tilde{b}\left(Y_s, \pi\right) - \tilde{b}\left(Y_s, \hat{\pi}\right)\|_2^2$$

$$\leq (C_{\tilde{b}} + \frac{1}{2}) \cdot \mathbb{E}\|X_s - Y_s\|_2^2 + \frac{1}{2}\|\tilde{b}(\cdot, \pi) - \tilde{b}(\cdot, \hat{\pi})\|_{2,\infty}^2;$$

and

$$(B) = \mathbb{E}\|\tilde{\sigma}\left(X_s, \pi\right) - \tilde{\sigma}\left(Y_s, \hat{\pi}\right)\|_F^2$$

$$\leq 2\mathbb{E}\|\tilde{\sigma}\left(X_s, \pi\right) - \tilde{\sigma}\left(Y_s, \pi\right)\|_F^2 + 2\mathbb{E}\|\tilde{\sigma}\left(Y_s, \pi\right) - \tilde{\sigma}\left(Y_s, \hat{\pi}\right)\|_F^2$$

$$\leq 2C_{\tilde{\sigma}}^2 \cdot \mathbb{E}\|X_s - Y_s\|_2^2 + 2\sup_x \|\tilde{\sigma}\left(x, \pi\right) - \tilde{\sigma}\left(x, \hat{\pi}\right)\|_F^2$$

$$:= 2C_{\tilde{\sigma}}^2 \cdot \mathbb{E}\|X_s - Y_s\|_2^2 + 2\|\tilde{\sigma}\left(\cdot, \pi\right) - \tilde{\sigma}\left(\cdot, \hat{\pi}\right)\|_{F,\infty}^2.$$

Combining the above, we get

$$\frac{\mathrm{d}}{\mathrm{d}s}\mathbb{E}\|X_s-Y_s\|_2^2 \leq \underbrace{(2C_{\tilde{b}}+1+2C_{\tilde{\sigma}}^2)}_{C_{\tilde{b},\tilde{\sigma}}}\mathbb{E}\|X_s-Y_s\|_2^2 + \underbrace{\|\tilde{b}(\cdot,\pi)-\tilde{b}(\cdot,\hat{\pi})\|_{2,\infty}^2 + 2\|\tilde{\sigma}(\cdot,\pi)-\tilde{\sigma}(\cdot,\hat{\pi})\|_{F,\infty}^2}_{C(\pi,\hat{\pi})}.$$

By Grönwall's inequality, we have

$$\mathbb{E}\|X_t-Y_t\|_2^2 \leq \frac{C(\pi,\hat{\pi})}{C_{\tilde{b},\tilde{\sigma}}}\left(e^{C_{\tilde{b},\tilde{\sigma}}t}-1\right). \tag{53}$$

Substituting back into (51), we obtain

$$W_2^2(\bar{d}_\mu^\pi, \bar{d}_\mu^{\hat{\pi}}) \leq \frac{C(\pi,\hat{\pi})}{C_{\tilde{b},\tilde{\sigma}}}\int_0^\infty \frac{1}{\beta}e^{-\beta s}\left(e^{C_{\tilde{b},\tilde{\sigma}}s}-1\right)\mathrm{d}s.$$

Thus, if $\beta > C_{\tilde{b},\tilde{\sigma}}$, we have

$$W_2(\bar{d}_\mu^\pi, \bar{d}_\mu^{\hat{\pi}}) \leq \frac{C(\pi,\hat{\pi})}{C_{\tilde{b},\tilde{\sigma}}(\beta - C_{\tilde{b},\tilde{\sigma}})\beta}.$$

Concerning the term $C(\pi,\hat{\pi})$, we have

$$\|\tilde{b}(\cdot,\pi)-\tilde{b}(\cdot,\hat{\pi})\|_{2,\infty} = \sup_x \|\tilde{b}(x,\pi)-\tilde{b}(x,\hat{\pi})\|_2 \leq \sup_x \|\hat{\pi}(\cdot|x)-\pi(\cdot|x)\|_1 \cdot \sup_{x,a}|b(x,a)|,$$

and

$$\|\tilde{\sigma}(\cdot,\pi)-\tilde{\sigma}(\cdot,\hat{\pi})\|_{F,\infty} = \sup_x \|\tilde{\sigma}(x,\pi)-\tilde{\sigma}(x,\hat{\pi})\|_F \leq \sqrt{n}\frac{\bar{\sigma}_0}{2\sigma_0}\sup_x \|\hat{\pi}(\cdot|x)-\pi(\cdot|x)\|_1^{\frac{1}{2}}.$$

Thus we have:

$$C(\pi,\hat{\pi}) = \|\tilde{b}(\cdot,\pi)-\tilde{b}(\cdot,\hat{\pi})\|_{2,\infty}^2 + 2\|\tilde{\sigma}(\cdot,\pi)-\tilde{\sigma}(\cdot,\hat{\pi})\|_{F,\infty}^2$$

$$\leq \left(\sup_{x,a}|b(x,a)|^2 + \frac{d\cdot\bar{\sigma}_0^2}{2\sigma_0^2}\right)\max\left(\sup_x \|\hat{\pi}(\cdot|x)-\pi(\cdot|x)\|_1, \sup_x \|\hat{\pi}(\cdot|x)-\pi(\cdot|x)\|_1^{\frac{1}{2}}\right)$$

which proves our upper bound. □

*Proof* (of Theorem 5). We have that

$$|\eta^{\hat{\pi}}-L^\pi(\hat{\pi})| = |\langle d_\mu^{\hat{\pi}} - d_\mu^\pi, f\rangle| = \frac{\|f\|_{\dot{H}^1}}{\beta}\left|\left\langle \bar{d}_\mu^{\hat{\pi}} - \bar{d}_\mu^\pi, \frac{f}{\|f\|_{\dot{H}^1}}\right\rangle\right|$$

$$\leq \frac{K}{\beta}\|\bar{d}_\mu^{\hat{\pi}} - \bar{d}_\mu^\pi\|_{\dot{H}^{-1}} \leq \frac{K\sqrt{M}}{\beta}W_2\left(\bar{d}_\mu^{\hat{\pi}}, \bar{d}_\mu^\pi\right). \tag{54}$$

where $K := \sup_{\hat{\pi}}\|f\|_{\dot{H}^1} < \infty$ (more about $K$ in the remarks below). Combining (54) with the estimate in (22) (of Lemma 4) yields the desired result in (23). □

*Remarks* (on $K$). In the performance-difference bound developed above, we assume $K$ is finite:

$$K := \|f\|_{\dot{H}^1} := \left(\int_{\mathbb{R}^n}|\nabla f(x)|^2\mathrm{d}x\right)^{\frac{1}{2}} < \infty,$$

where $f(x;\pi,\hat{\pi}) := \int_{\mathcal{A}}\hat{\pi}(a\mid x)\left(q(x,a;\pi)+p(x,a,\hat{\pi})\right)\mathrm{d}a$. The famous Poincaré inequality can provide a lower bound on this quantity; but we need an upper bound as well, i.e.,

$$K = \left(\int_{\mathbb{R}^n}|\nabla f(x)|^2\mathrm{d}x\right)^{\frac{1}{2}} \leq C\left(\int_{\mathbb{R}^n}|f(x)|^2\mathrm{d}x\right)^{\frac{1}{2}}.$$

This above is essentially a *reverse* Poincaré Inequality, which is not likely to hold (in particular, the existence of the constant $C$).

Should we indeed have a reverse Poincaré Inequality, then we can further bound $f$ by

$$|f(x)| = |\int_{\mathcal{A}} (\hat{\pi}(a \mid x) - \pi(a \mid x)) (q(x, a; \pi) + p(x, a, \hat{\pi})) \, da|$$

$$\leq \int_{\mathcal{A}} |\hat{\pi}(a \mid x) - \pi(a \mid x)| \cdot |q(x, a; \pi) + p(x, a, \hat{\pi})| \, da$$

$$\leq 2 \sup_a |q(x, a; \pi) + p(x, a, \hat{\pi})| \, D_{\mathrm{TV}}(\pi(\cdot \mid x), \hat{\pi}(\cdot \mid x)),$$

and

$$\left( \int_{\mathbb{R}^n} |f(x)|^2 dx \right)^{\frac{1}{2}} \leq \left( \int_{\mathbb{R}^n} 4 \sup_a |q(x, a; \pi) + p(x, a, \hat{\pi})|^2 \, D_{\mathrm{TV}}^2(\pi(\cdot \mid x), \hat{\pi}(\cdot \mid x)) dx \right)^{\frac{1}{2}}$$

$$\leq \left( \int_{\mathbb{R}^n} 2 \sup_a |q(x, a; \pi) + p(x, a, \hat{\pi})|^2 \, dx \right)^{\frac{1}{2}} \sqrt{\sup_x D_{\mathrm{KL}}(\pi(\cdot \mid x), \hat{\pi}(\cdot \mid x))},$$

where the second inequality is from Pinsker's inequality. This way, we would have recovered a similar bound as in the discrete RL. Since we do not have the reverse Poincaré inequality, however, we have to assume that $K$ is finite.

## Appendix C  Algorithms

### C.1  Performance of CPPO with Square-root KL and Linear KL

Here we present a detailed version of the CPPO algorithm. For two probability distributions $P$ and $Q$ over the action space with density functions $p$ and $q$ correspondingly, recall that the KL-divergence between these two is defined as:

$$D_{\mathrm{KL}}(P \| Q) = \int_{\mathcal{A}} \log(\frac{q(a)}{p(a)}) q(a) da,$$

Denote $D_{\mathrm{KL}}(\theta \| \theta_k) := \mathbb{E}_{x \sim d_\mu^{\theta_k}} D_{\mathrm{KL}}(\pi_{\theta_k}(\cdot | x) \| \pi_\theta(\cdot | x))$, to distinguish it from $\bar{D}_{\mathrm{KL}}(\theta \| \theta_k) := \mathbb{E}_{x \sim d_\mu^{\theta_k}} \sqrt{D_{\mathrm{KL}}(\pi_{\theta_k}(\cdot | x) \| \pi_\theta(\cdot | x))}$ which was used in CPPO Algorithm in 2.

Note that bounding the performance difference by the linear KL-divergence $D_{\mathrm{KL}}(\theta, \theta_k)$, instead of its square-root counterpart $\bar{D}_{\mathrm{KL}}(\theta \| \theta_k)$, will generally require stronger conditions (which may be difficult to satisfy). For completeness, we present the following algorithm as an important benchmark, the CPPO with linear KL-divergence:

---

**Algorithm 3** CPPO: PPO with adaptive penalty constant (linear KL-divergence)

---

**Input**: Policy parameters $\theta_0$, critic net parameters $\phi_0$

1: **for** $k = 0, 1, 2, \cdots$ until $\theta_k$ converge **do**

2:     Collect a truncated trajectory $\{X_{t_i}, a_{t_i}, r_{t_i}, p_{t_i}\}, i = 1, \ldots, N$ from the environment using $\pi_{\theta_k}$.

3:     **for** $i = 0, \ldots, N - 1$ **do**: Update the critic parameters as in (8)

4:     **for** $j = 1, , \ldots, J$ **do**: Draw i.i.d. $\tau_j$ from $\exp(\beta)$, round $\tau_j$ to the largest multiple of $\delta_t$ no larger than it, and compute the GAE estimator of $q(X_{\tau_j}, a_{\tau_j})$

$$\tilde{q}(X_{\tau_j}, a_{\tau_j}) := \left( r_{\tau_j} \delta_t + e^{-\beta \delta_t} V(X_{\tau_j + \delta_t}) - V(X_{\tau_j}) \right) / \delta_t.$$

5:     Compute policy update (by taking a fixed $s$ steps of gradient descent)

$$\theta_{k+1} = \arg\max_\theta L^{\theta_k}(\theta) - C_{\mathrm{penalty}}^k D_{\mathrm{KL}}(\theta \| \theta_k).$$

6:     **if** $D_{\mathrm{KL}}(\theta_{k+1} \| \theta_k) \geq (1 + \epsilon)\delta$, **then**     $C_{\mathrm{penalty}}^{k+1} = 2 C_{\mathrm{penalty}}^k$.

7:     **else if** $D_{\mathrm{KL}}(\theta_{k+1} \| \theta_k) \leq \delta/(1 + \epsilon)$, **then**     $C_{\mathrm{penalty}}^{k+1} = C_{\mathrm{penalty}}^k / 2$.

---

A comparison between the above and Algorithm 2 (using square-root KL divergence) is presented in §D.3 below, which clearly illustrates the advantage of square-root KL divergence.

## C.2 KL-divergence

We elaborate here on the KL-divergence between the current policy and the optimal policy, along with the entropy regularizer. By the performance difference formula, we have

$$\eta(\pi) - \eta(\pi^*) = \int_{\mathbb{R}^n} d_\mu^\pi(x) \left[ \int_{\mathcal{A}} \pi(a \mid x) \left(q(x, a; \pi^*) - \gamma \log(\pi(a))\right) \mathrm{d}a \right] \mathrm{d}x.$$

Notice that by the definition of KL-divergence we defined before, we have

$$D_{\mathrm{KL}}(\pi^*(\cdot|x)\|\pi(\cdot|x)) = \int_{\mathcal{A}} \log(\frac{\pi(a|x)}{\pi^*(a|x)})\pi(a|x)\mathrm{d}a.$$

Similar as the previous discussion of soft $q$-learning, $\pi^*$ is optimal implies that

$$\pi^*(a \mid x) \propto \exp(\frac{q(x, a, \pi^*)}{\gamma}),$$

and the normalization constant is 1 can be proved through considering the exploratory HJB equation, see [22, 56]. Thus

$$D_{\mathrm{KL}}(\pi^*(\cdot|x)\|\pi(\cdot|x)) = \int_{\mathcal{A}} \log(\pi(a|x))\pi(a|x)\mathrm{d}a - \int_{\mathcal{A}} \frac{q(x, a, \pi^*)}{\gamma}\pi(a|x)\mathrm{d}a,$$

which leads to

$$\eta(\pi) - \eta(\pi^*) = -\gamma \cdot \mathbb{E}_{x \sim d_\mu^\pi} D_{\mathrm{KL}}(\pi^*(\cdot|x)\|\pi(\cdot|x)).$$

This justifies our claim that the KL-divergence is essentially equivalent to the distance to the optimal performance.

# Appendix D   Experiments

## D.1   Example 1

Recall, in the LQ control problem, the reward function is

$$r(x, a) = -\left(\frac{M}{2}x^2 + Rxa + \frac{N}{2}a^2 + Px + Qa\right),$$

with $M \geq 0, N > 0, R, Q, P \in \mathbb{R}$ and $R^2 < MN$, and we adopt the entropy regularizor as

$$p(x, a, \pi) = -\log(\pi(a)).$$

Furthermore, suppose that the discount rate satisfies $\beta > 2A + C^2 + \max\left(\frac{D^2 R^2 - 2NR(B+CD)}{N}, 0\right)$.

The following results are readily derived from Theorem 4 of [58]. The value function of the optimal policy $\pi^*$ is

$$V(x) = \frac{1}{2}k_2 x^2 + k_1 x + k_0, \quad x \in \mathbb{R},$$

where

$$k_2 := \frac{1}{2} \frac{\left(\rho - (2A + C^2)\right) N + 2(B + CD)R - D^2 M}{(B + CD)^2 + (\rho - (2A + C^2)) D^2}$$

$$-\frac{1}{2} \frac{\sqrt{\left((\rho - (2A + C^2)) N + 2(B + CD)R - D^2 M\right)^2 - 4\left((B + CD)^2 + (\rho - (2A + C^2)) D^2\right)(R^2 - MN)}}{(B + CD)^2 + (\rho - (2A + C^2)) D^2},$$

$$k_1 := \frac{P\left(N - k_2 D^2\right) - QR}{k_2 B(B + CD) + (A - \rho)\left(N - k_2 D^2\right) - BR},$$

and

$$k_0 := \frac{(k_1 B - Q)^2}{2\rho\left(N - k_2 D^2\right)} + \frac{\gamma}{2\rho}\left(\ln\left(\frac{2\pi e\gamma}{N - k_2 D^2}\right) - 1\right)$$

respectively. Moreover, the optimal feedback control is Gaussian, with density function

$$\pi^*(a;x) = \mathcal{N}\left(a \mid \frac{(k_2(B+CD) - R)\,x + k_1 B - Q}{N - k_2 D^2}, \frac{\gamma}{N - k_2 D^2}\right).$$

For a set of model parameters: $A = -1, B = C = 0, D = 1, M = N = Q = 2, R = P = 1, \beta = 1, \gamma = 0.1$, following the formulas and the parameterized policy $\pi_\theta(\cdot \mid x) = \mathcal{N}(\theta_1 x + \theta_2, \exp(\theta_3))$, and the corresponding value function $V_\phi(x) = \frac{1}{2}\phi_2 x^2 + \phi_1 x + \phi_0$, we can derive the optimal parameters:

$$\phi^* = [0.71914874, -0.10555128, -0.53518376],$$

and

$$\theta^* = [-0.39444872, -0.78889745, -1.40400944].$$

Table 1: Hyper-parameter values for Example 1

| Alphabet | Description | Value |
|---|---|---|
| $T$ | Trajectory Truncation Length | 25 |
| $\beta$ | discount factor | 1 |
| $\delta_t$ | time interval | 0.005 |
| $J$ | batch size for sampling $\exp(\beta)$ | 100 |
| $\alpha_1$ | learning rate for policy iteration $k$ | 0.02 when $k \le 50$ and $0.02 \times \log(\frac{50}{k})$ when $k > 50$ |
| $\alpha_2$ | learning rate for value iteration $k$ | 0.01 when $k \le 50$ and $0.01 \times \log(\frac{50}{k})$ when $k > 50$ |
| $K$ | iteration threshold | 2000 |
| $s$ | steps of gradient descent | 10 |
| $\delta$ | radius | 0.0002 |
| $\epsilon$ | tolerance level | 0.5 |

## D.2   Example 2

The model parameters are $k = 0.01, \theta = 7, \eta = 0.1, \rho = 0.3, \sigma = 1, r_f = 0.01, \ell = 5$. For both the value function and the policy parameterization, we use a 3-layer neural network, and with the initial parameters sampled form the uniform distribution over [-0.5,0.5]. We use the tanh activation function for the hidden layer.

Table 2: Hyperparameter values for Example 2

| Alphabet | Description | Value |
|---|---|---|
| $T$ | Trajectory Truncation Length | 25 |
| $\beta$ | discount factor | 1 |
| $\delta_t$ | time interval | 0.005 |
| $J$ | batch size for sampling $\exp(\beta)$ | 100 |
| $\alpha_1$ | learning rate for policy iteration $k$ | 0.005 when $k \le 50$ and $0.005 \times \log(\frac{50}{k})$ when $k > 50$ |
| $\alpha_2$ | learning rate for value iteration $k$ | 0.01 when $k \le 50$ and $0.01 \times \log(\frac{50}{k})$ when $k > 50$ |
| $K$ | iteration threshold | 200 |
| $s$ | steps of gradient descent | 10 |
| $\delta$ | radius | 0.025 |
| $\epsilon$ | tolerance level | 0.5 |

## D.3   Performance of CPPO with Square-root KL and Linear KL

We compare the performance of CPPO with square-root KL-divergence (denote as CPPO), and linear KL-divergence (denoted as CPPO (nst) — non square-root) applied to the experiments in Example 1 and Example 2.  Figure 4 compares the distance between the current policy parameters and the optimal parameters, with $x$-axis denoting the iteration times and $y$-axis denoting the $L_2$ distance. Figure 5 compares the current expected return, with $x$-axis denoting the iteration times and $y$-axis denoting the current performance by taking the average of 100 times of Monte Carlo evaluation. In

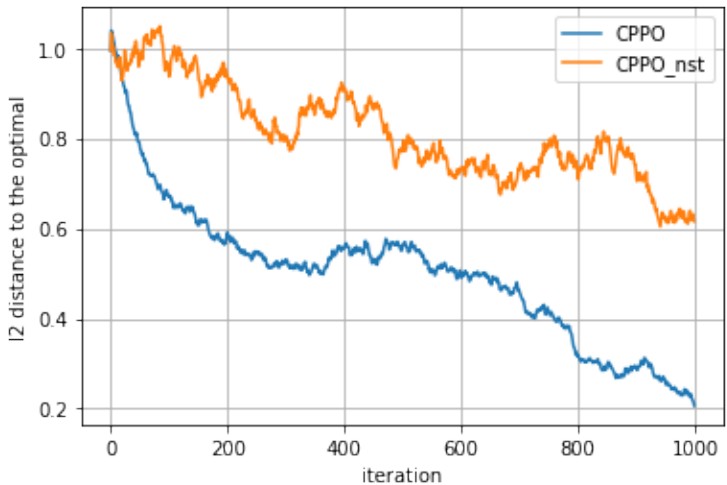

Figure 4: Performance of CPPO and CPPO (nst) to the Example 1

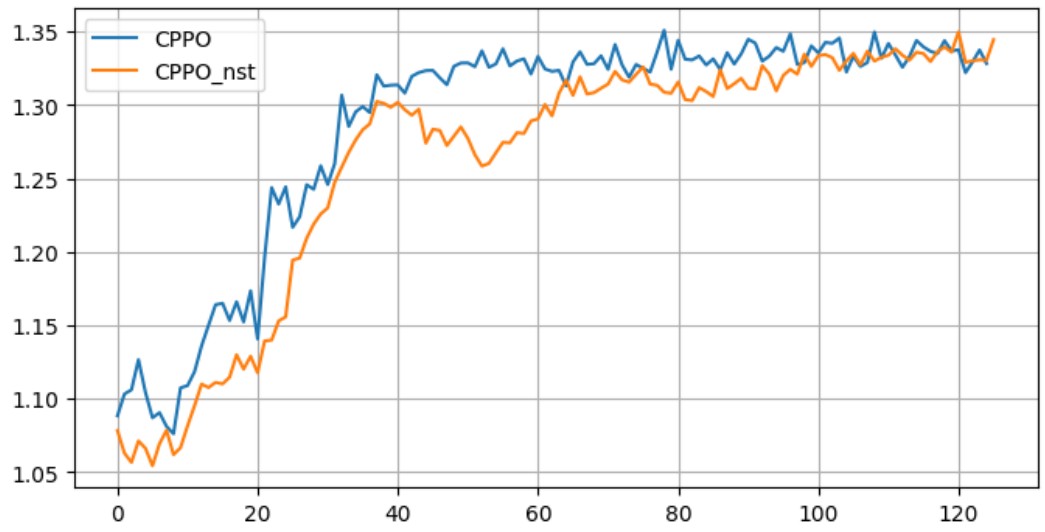

Figure 5: Performance of CPPO and CPPO (nst) to the Example 2

both figures, the blue curve represents the algorithm with square-root KL-divergence as opposed to the orange one corresponding to the linear version. Both figures clearly demonstrate the advantage of the former. In particular, the linear version can suffer from getting stuck at the local optimum as demonstrated in Example 1.

### D.4  Performance of CPG and CPPO compared to the classical discrete-time algorithms

We conduct experiments to compare the CPG and CPPO to their discrete counterparts. Specifically, we discretize the MDP in Example 1, and implement the classical PG and PPO algorithms. Our results show that in time discretization with step size $\delta t = 0.1$ and $\delta t = 0.05$, the performance of CPG and CPPO is (at least) comparable to their discrete counterparts; in particular, for $\delta t = 0.1$, CPG outperforms PG. We have repeated the experiments for 25 random seeds, and plotted both the average performance line and the error bar. These experimental results indicate that the continuous approach has the potential to outperform their discrete counterparts, which is worth further exploring in the future.

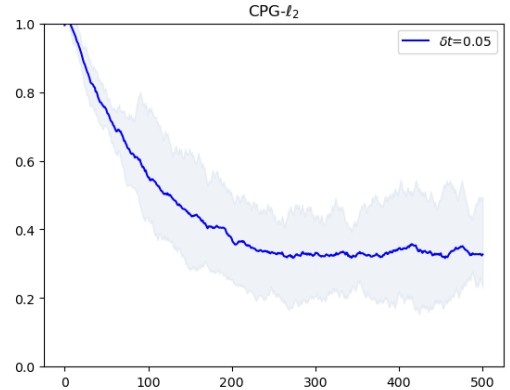

Figure 6: CPG in $l_2$ distance ($\delta_t = 0.05$)

Figure 7: CPG in KL distance ($\delta_t = 0.05$)

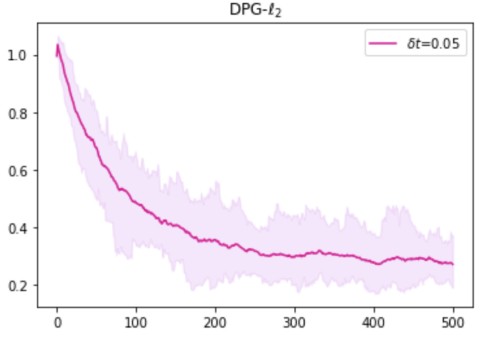

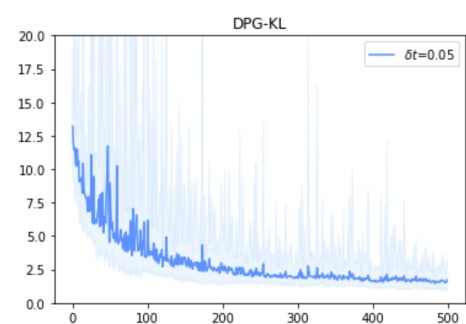

Figure 8: DPG in $l_2$ distance ($\delta_t = 0.05$)

Figure 9: DPG in KL distance ($\delta_t = 0.05$)

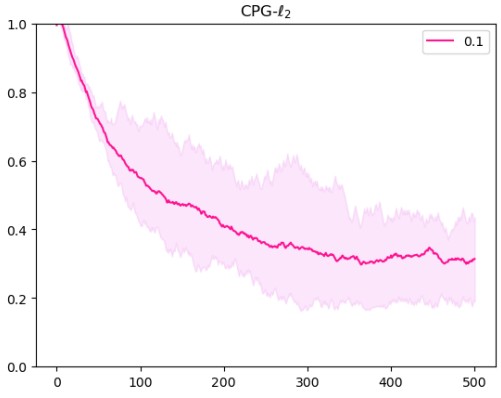

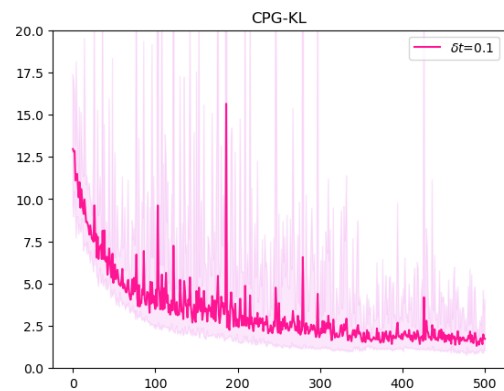

Figure 10: CPG in $l_2$ distance ($\delta_t = 0.1$)

Figure 11: CPG in KL distance ($\delta_t = 0.1$)

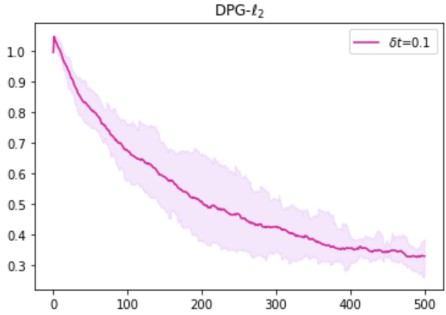

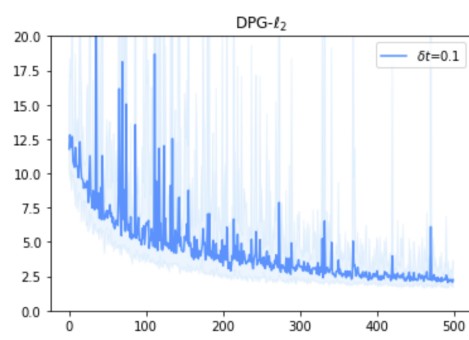

Figure 12: DPG in $l_2$ distance ($\delta_t = 0.1$)

Figure 13: DPG in KL distance ($\delta_t = 0.1$)

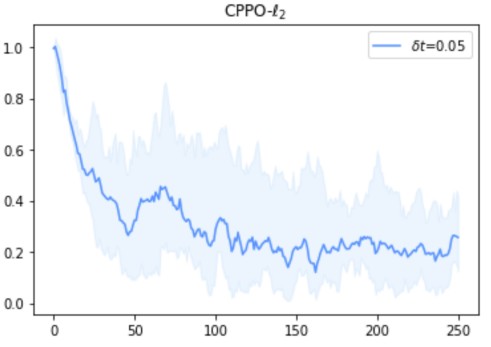

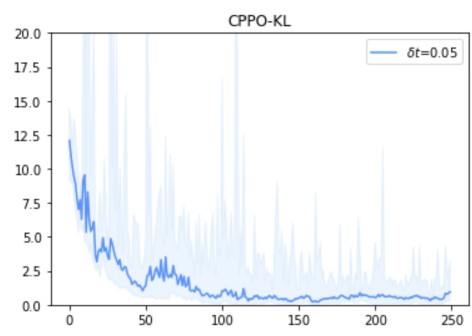

Figure 14: CPPO in $l_2$ distance ($\delta_t = 0.05$)

Figure 15: CPPO in KL distance ($\delta_t = 0.05$)

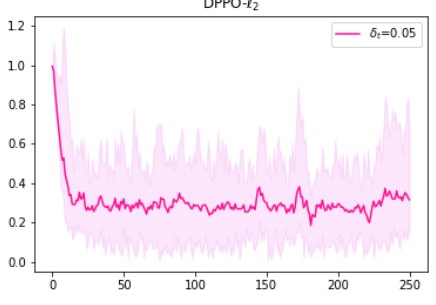

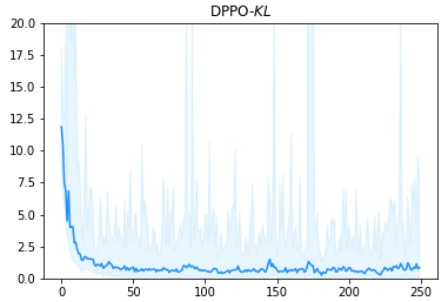

Figure 16: DPPO in $l_2$ distance ($\delta_t = 0.05$)

Figure 17: DPPO in KL distance ($\delta_t = 0.05$)

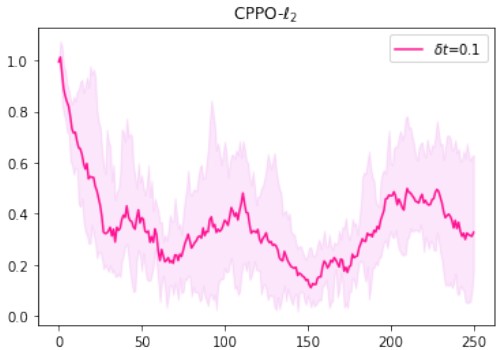

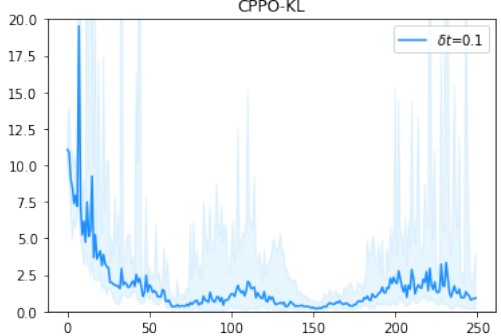

Figure 18: CPPO in $l_2$ distance ($\delta_t = 0.1$)    Figure 19: CPPO in KL distance ($\delta_t = 0.1$)

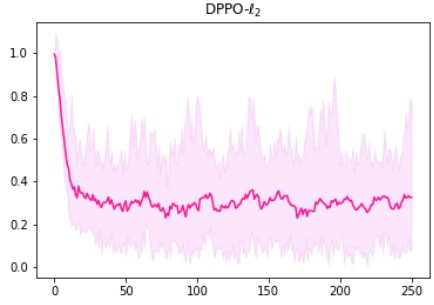

Figure 20: DPPO in $l_2$ distance ($\delta_t = 0.1$)    Figure 21: DPPO in KL distance ($\delta_t = 0.1$)

