# OpenReview forum: "Policy Optimization for Continuous Reinforcement Learning"
_NeurIPS.cc/2023/Conference — NeurIPS 2023 poster_

### Official Review · Reviewer_vuEi · 2023-07-02

**Soundness:** 2 fair
**Presentation:** 3 good
**Contribution:** 3 good
**Rating:** 5
**Confidence:** 3

**Summary:**

Reinforcement learning (RL) is a powerful tool for solving sequential decision making problems but has primarily been formulated for discrete-time Markov decision processes. However, many real-world systems are more naturally expressed in continuous time, and the proper choice of discretization time step may be challenging. Additionally, a controller operating in continuous time may be more suitable for high frequency applications. Prior work has formulated a continuous-time analog to the advantage function, called the q-value. Building on the concept of a q-value, the paper sets out to derive a continuous-time analog to the discounted occupation probability and performance difference lemma, two core concepts in RL. The authors then show how to compute a gradient of this performance difference lemma to give a continuous analog to the policy gradient. Next, the authors form a local approximation of the performance difference which takes an expectation over the current policy rather than the updated one. They provide bounds on the gap between the true performance metric and this local approximation. Then they use this local approximation to formulate a continuous analog to TRPO\PPO. The paper evaluates the continuous-time policy gradient and PPO algorithms on synthetic linear-quadratic stochastic control problems. They also consider a two-dimensional optimal pair trading problem and show their CPPO algorithm performs best.

**Strengths:**

- Continuous-time RL is an under-explored area that has application in controlling systems at high frequency and using an adaptive discretization scheme with non-uniform time steps.
- This appears to be the first work to propose a continuous-time analog in a stochastic setting to the performance difference lemma and derive a continuous PPO algorithm.
- The paper is well organized and clearly written. It does a good job explaining the results and provides enough information to support its claims.

**Weaknesses:**

- There is no discussion or experimental results which illustrate how the continuous-time formulation is advantageous over discrete-time RL. The proposed algorithms still ultimately require us to discretize. As such, there should be some comparison to existing discrete-time RL methods. The hope would be that formulating the problem in continuous time allows us to achieve better performance under certain choices of step size. Another potential benefit of continuous-time RL is the case of unevenly-spaced observations. Experiments which highlight these benefits would significantly strengthen the paper.
- The problems considered are toy examples which do not tell us much about the scalability of the proposed methods. More complex scenarios would make the paper much more convincing. Especially ones in which the continuous-time formulation provides a clear advantage.
- The discussion of related work could be better. It is still a little unclear to me how this work is positioned in the continuous-time RL literature.

**Questions:**

- How do these continuous-time methods compare to their discrete-time counterparts when the time step discretization is uniform? How does the performance gap change with choice of discretization time step? What are the main advantages of this formulation?
- How does this approach scale to harder, more realistic problems?
- How exactly does the continuous-time policy gradient algorithm in this paper compare to previously proposed approaches?

**Limitations:**

There is no discussion of limitations of the continuous-time formulation or their specific algorithms. The paper would be stronger if it discussed these. This could include assumptions made in the proofs or the fact that the work assumes the dynamics follow an Ito SDE driven by Brownian motion.

---

> ### Author Rebuttal · Authors · 2023-08-08
>
> We thank the reviewer for the detailed comments, especially on the experiments.
>
> 1. There is no discussion or experimental results which illustrate how the continuous-time formulation is advantageous...
>
> A: We have conducted extra experiments to compare the CPG and CPPO to their discrete counterparts. (The results will be included in the revision). Specifically, we discretize the MDP in Example 1, and implement the classical PG and PPO algorithms. Our results show that in time discretization with step size $\delta t=0.1$ and $\delta t= 0.05$, the performance of CPG and CPPO is (at least) comparable to their discrete counterparts; in particular, for $\delta t=0.1$, CPG outperforms PG. We have repeated the experiments for 25 random seeds, and plotted both the average performance line and the error bar. Please refer to the link https://www.dropbox.com/scl/fo/03g1ub7mvis64yucclqzm/h?rlkey=boyq188kpop55hj5ahyiyom16&dl=0. These experimental results indicate that the continuous approach has the potential to outperform their discrete counterparts. While this is quite preliminary, we do plan to do more thorough investigations along this line in the near future.
>
> We would like to emphasize that our main objective (and contribution) is to provide a continuous approach (which appears under-developed), with rigorous analyses and provable results, both of which appear to be difficult to do in the discrete setting. On the numerical side, preliminary experimentation (as summarized above) has shown that the continuous approach is at least comparable to (or outperform) its discrete counterpart.
>
>
> 2. More complex scenarios would make the paper much more convincing...
>
> A: As mentioned above (in point 1), our focus here is to develop a continuous approach for policy optimization, supported by rigorous analyses and provable results. While numerical studies have shown promising performance of the continuous approach, as highlighted above in point 1, we agree that these are preliminary and limited in scope, and more thorough experimentation is needed.
>
> We believe that our proposed algorithms can be used in many scenarios (financial trading, aviation control, etc) involving large-scale and high-frequency data. It will certainly warrant another independent study (which we do plan to pursue as a next step) to implement our proposed algorithms and evaluate their performance in one of such applications.
>
>
> 3. The discussion of related work could be better.
>
> A: To the best of our knowledge, this is the first paper that studies policy optimization in the continuous and stochastic setting. While this is a notably under-developed area in RL, we do have cited (on p.1-2) several related works and discussed how do they motivate or relate to our work.
>
> 4. How does the performance gap change with the choice of discretization time step...
>
> A: We have implemented our proposed algorithms with different (time) step sizes: $\delta t=0.02$, $\delta t=0.05$ and $\delta t=0.1$, and found that the performances are quite similar, suggesting that our CPG algorithm is insensitive to step size; please refer to the link https://www.dropbox.com/scl/fo/mkuqf1nux1ysiwhys9oqn/h?rlkey=8s4ui8i2wknd4za1o11alf1u5&dl=0 for details.
>
> 5. Limitation of the Ito-SDE formulation...
>
> A: Similar to the classical MDP setting, we need to assume a Markovian and stationary setting in our continuous approach, which is represented by an It\^o-SDE. Other more technical conditions are standard (and very mild), such as requiring the It\^o-SDE to be well-posed, the continuity and growth condition on model parameters. In the revision, we will state these conditions more explicitly.

---

### Official Review · Reviewer_9Un8 · 2023-07-05

**Soundness:** 2 fair
**Presentation:** 3 good
**Contribution:** 3 good
**Rating:** 5
**Confidence:** 3

**Summary:**

This paper investigates the continuous reinforcement learning problem. The proposed method is based on the notion of occupation time for policy gradient, which is analogous to the visitation frequency in discrete Markov decision processes. Empirical evaluations are conducted on two example scenarios for two versions of the algorithm (CPG and CPPO).

**Strengths:**

- Analyze an Important formulation of continuous time and space in reinforcement learning.
- Theoretical analysis is present to develop a policy gradient counterpart of TRPO/PPO.
- The empirical evaluation is presented along with theoretical analysis.

**Weaknesses:**

- Lack of comparison with discrete counterpart.
- The lack of implementation details (missing hyperparameter, random seed) makes reproducibility challenging.
- The evaluation of the proposed method is limited to only two hand-crafted examples, which hinders a comprehensive understanding of its implications.

**Questions:**

In Figures 1 and 2, what does it mean to be the distance to the optimal to be some non-zero value? How does this translate to the reward performance? Reward performance is critical to know whether or not the achieved difference in distance is useful.

The performance is compared with the proposed two methods, CPG and CPPO (Figure 3), and no baseline is considered. An essential choice would be to use the discrete counterpart (Policy Gradient - PG and PPO). A simple form can be to make the MDP discrete and run PG and PPO. This comparison is even important to understand what kind of problem the continuous RL is useful for.

Is the achieved return by CPPO optimal (empirical best) in Figure 3?

Implementation details need to be included; what are the PPO-specific hyperparameters (e.g., clipping), for Example 2?

For reproducibility, a common practice is using several seed runs to account for variation in the results. How many random seeds are used for the experiments in Figures 1 and 2? These missing details make the presented results hard to reproduce.

**Limitations:**

Yes.

---

> ### Author Rebuttal · Authors · 2023-08-08
>
> We thank the reviewer for the detailed comments, especially on the experiments.
>
> 1. Lack of comparison to the discrete counterpart.
>
> A: We have conducted extra experiments to compare the CPG and CPPO to their discrete counterparts. (The results will be included in the revision). Specifically, we discretize the MDP in Example 1, and implement the classical PG and PPO algorithms. Our results show that in time discretization with step size $\delta t=0.1$ and $\delta t= 0.05$, the performance of CPG and CPPO is (at least) comparable to their discrete counterparts; in particular, for $\delta t=0.1$, CPG outperforms PG. We have repeated the experiments for 25 random seeds, and plotted both the average performance line and the error bar. Please refer to the link https://www.dropbox.com/scl/fo/03g1ub7mvis64yucclqzm/h?rlkey=boyq188kpop55hj5ahyiyom16&dl=0. These experimental results indicate that the continuous approach has the potential to outperform their discrete counterparts. While this is quite preliminary, we do plan to do more thorough investigations along this line in the near future.
>
> We would like to emphasize that our main objective (and contribution) is to provide a continuous approach (which appears under-developed), with rigorous analyses and provable results, both of which appear to be difficult to do in the discrete setting. On the numerical side, preliminary experimentation (as summarized above) has shown that the continuous approach is at least comparable to (or outperform) its discrete counterpart.
>
> 2. The lack of implementation details.
>
> A: On p.8, l.279, we have referred the implementation details to Appendix D. In addition, we plan to post our codes on Github, so the reproducibility will not be an issue. As to the random seeds, we use 25 random seeds to run the algorithms. More details will be added to the revision.
>
> 3. The evaluation of the proposed method is limited to...
>
> A: As mentioned above (in point 1), our focus here is to develop a continuous approach for policy optimization, supported by rigorous analyses and provable results. While numerical studies have shown promising performance of the continuous approach, as highlighted above in point 1, we agree that these are quite preliminary and limited in scope, and more thorough experimentation is needed, which we plan to do in the near future.
>
> Indeed we believe that our proposed algorithms can be used in many scenarios (financial trading, aviation control, etc) involving large-scale and high-frequency data. It will certainly warrant another independent study to implement our proposed algorithms and evaluate their performance in one of such applications.
>
> 4. Connection of policy distance and reward performance difference
>
> A: As we mentioned on p.9, l.282-28, minimizing the KL-divergence (between the iterated policy and the optimal policy)
> is equivalent to minimizing the distance between the current policy objective and the optimal objective; and we referred to Appendix D.1 for further details. In addition, preliminary experiments indicate that our proposed algorithms do converge to the (local) optimum.
>
> 5. The performance is compared with the proposed two methods...
>
> A: We have now conducted extra experiments and made more comparisons as summarized in point 1.
>
> 6. Implementation details need to be included; what are the PPO-specific hyperparameters (e.g., clipping), for Example 2?
>
> A: We did not use the clipping technique. Details regarding the hyperparameter are spelled out in Appendix D (as mentioned  on p.9,  l. 297).

---

> > ### Comment · Reviewer_9Un8 · 2023-08-20
> >
> > I would like to thank the authors for their detailed responses and for conducting additional experiments. Based on the updated results and content, I have raised my rating from 4 to 5.

---

### Official Review · Reviewer_Mbd6 · 2023-07-07

**Soundness:** 3 good
**Presentation:** 3 good
**Contribution:** 3 good
**Rating:** 7
**Confidence:** 3

**Summary:**

This paper seeks to answer three research questions: 1) Is there a continuous time analog of the state occupancy measure, 2) Is there a convenient expression for the performance difference between two policies in the continuous time setting, and 3) Can PPO be adapted to fit the continuous time setting? In answering these questions, the paper presents a continuous time occupancy measure, a policy performance difference similar to that of conservative policy iteration and related algorithms, and continuous versions of REINFORCE and PPO are provided. Theoretical results for several properties are given, and experiments demonstrating that the algorithms were able to solve a couple of problems are provided.


**Strengths:**

The paper provides a thorough theoretical treatment of defining policy gradients for the continuous time RL setting. The paper clearly defines and adequately answers its stated objectives.

**Weaknesses:**

The biggest area for improvement in this paper is in the empirical results. While the main results of this paper are, theoretical new practical algorithms are presented. Thus, they deserve proper evaluation and experimentation to educate the reader on the challenges of using them.

For example, there are no experiments illustrating that there were any special difficulties in applying these algorithms to the continuous setting. There should be experiments illustrating how the hyperparameters, particularly those specific to the continuous time setting, impact the optimization process. Currently, the results only tell us that the algorithms were made to work. Great, but we do not learn anything beyond this trivial result. It is important to develop the reader’s understanding of how they can make the algorithms work or at least what can cause failure.


Additionally, the adaptive penalty term should also be explored since it deviates from the standard PPO implementation. Does it effectively constrain the distribution throughout learning? How does the penalty change over time? These are important questions to answer when introducing a new method.


Minor quibble: the opening sentence of the introduction is boring. Many papers have used this form, and my eyes glaze over as soon as I read them. Try keeping this motivation brief and tie it immediately to the scope of the work discussed in this paper.

**Questions:**

- How does the sampling at discretization impact the gradient estimate?

- How does the step size need to be adapted to work in the continuous-time setting? Is it sensitive to the scaling of the time horizon or sampling rate?

---

> ### Author Rebuttal · Authors · 2023-08-08
>
> We thank the reviewer for the suggestions and comments.
>
> 1. How does the sampling at discretization impact the gradient estimate?
>
> A: We have implemented our proposed algorithms with different (time) step sizes: $\delta t=0.02$, $\delta t=0.05$ and $\delta t=0.1$, and found that the performances are quite similar, suggesting that our CPG algorithm is insensitive to step size; please refer to the link https://www.dropbox.com/scl/fo/mkuqf1nux1ysiwhys9oqn/h?rlkey=8s4ui8i2wknd4za1o11alf1u5&dl=0 for details.
>
> 2. Comparison of our proposed algorithm in continuous setting with the discrete MDP algorithms.
>
> A:  We have conducted extra experiments to compare the CPG and CPPO to their discrete counterparts. (The results will be included in the revision). Specifically, we discretize the MDP in Example 1, and implement the classical PG and PPO algorithms. Our results show that in time discretization with step size $\delta t=0.1$ and $\delta t= 0.05$, the performance of CPG and CPPO is (at least) comparable to their discrete counterparts; in particular, for $\delta t=0.1$, CPG outperforms PG. We have repeated the experiments for 25 random seeds, and plotted both the average performance line and the error bar. Please refer to the link https://www.dropbox.com/scl/fo/03g1ub7mvis64yucclqzm/h?rlkey=boyq188kpop55hj5ahyiyom16&dl=0. These experimental results indicate that the continuous approach has the potential to outperform their discrete counterparts. While this is quite preliminary, we do plan to do more thorough investigations along this line in the near future.
>
> 3. How does the step size need to be adapted to work in the continuous-time setting?
>
> A: As mentioned above, the performance of our algorithms appears to be quite robust to step size. In the future, we plan to also investigate the sensitivity of the hyperparameters to the environment dynamics (e.g., the time horizon).

---

> > ### Comment · Reviewer_Mbd6 · 2023-08-14
> > **Response to author response**
> >
> > The plots are not useful for comparing since they are all on different graphs.
> >
> > >As mentioned above, the performance of our algorithms appears to be quite robust to step size.
> >
> > What evidence do you have for this? It is not evident to me that one would expect the step size to be robust to changes in time discretization.

---

> > > ### Author Response · Authors · 2023-08-15
> > >
> > > Many thanks for your further questions and comments.
> > >
> > > For the plots, we will add the new graphs directly evaluating policy in the revised version, to allow a more precise comparison of the algorithm performance. Currently, because of time constraints, we added two new graphs, named "CPG-KL" and "CPG_l2", by concatenating the plots of different time discretization into one figure, for KL-divergence and l-2 distance separately, and we believe that the plots of KL-divergence could justify our claim as the performance of the CPG algorithm appears to be similar for different time discretization from our experiment results. Please see the link https://www.dropbox.com/scl/fo/mkuqf1nux1ysiwhys9oqn/h?rlkey=8s4ui8i2wknd4za1o11alf1u5&dl=0 for updated contents.
> > >
> > > For the step size issue, we agree that we need further experiments to test the robustness of the algorithm's performance to different choices of step size.  We will also investigate how to quantitatively build the connection between the suggested / optimal learning rate (step size) and the model parameters (e.g. sampling rate, time horizon) in future works.

---

### Decision · Program_Chairs · 2023-09-21

**Decision:**

Accept (poster)

**Comment:**

The reviewers all praised this work for the quality of its presentation and the strength of its contribution, with many commenting on the novelty and importance of this analysis in the continuous setting. All of the reviewers initially critiqued the empirical results, calling for further evaluation and comparison with discrete counterparts. The authors' rebuttal resolved these concerns, and consequently, the merits of this paper far outweigh the remaining weaknesses, and as such I recommend acceptance.